# Compute Optimal Tokenization

## Abstract

Scaling laws enable the optimal selection of data amount and language model size, yet the impact of the data unit, *the token*, on this relationship remains underexplored. In this work, we systematically investigate how the information granularity of tokens, controlled by the compression rate (i.e., average bytes of text per token), affects scaling trends. We train 988 latent tokenized models (BLT) ranging from 50M to 7B parameters that enable setting the desired compression rate. This flexibility allows us to study the role of compression rate well beyond 4.57 bytes per token obtained with a popular BPE tokenizer. Our experiments reveal that in compute-optimal configurations, model parameter counts scale proportionally to data size measured in *bytes*, not in *tokens* as commonly perceived (Kaplan et al., 2020; Hoffmann et al., 2022). Furthermore, we discover that the optimal compression rate differs from the one obtained with BPE and decreases with compute. These findings generalize to both latent and subword tokenization, as well as to languages other than English, guiding language model developers on tokenization scheme selection for maximal compute efficiency.

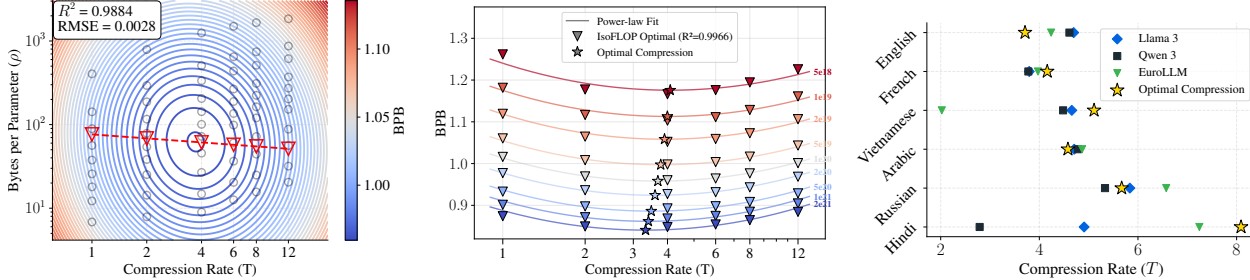

(a) Optimal bytes per parameter ratio across compression rates. Fixed training budget $10^{20}$ FLOPs.

(b) Optimal compression rate based on a scaling law fit. Training budget marked by color.

(c) Optimal compression rate differs from compression of subword tokenizers

Figure 1: Key findings of this work: 1a in compute optimal scaling: bytes (not tokens) of data increase proportionally to parameter count; 1b for each training budget and dataset, we find optimal compression rate, its value decreases with scale; 1c the optimal compression rate varies across languages and differs from compression of popular BPE tokenizers.

## 1 Introduction

Scaling laws have informed the efficient design of language models, prescribing the optimal balance between model size and training data (Kaplan et al., 2020; Hoffmann et al., 2022). Standard approaches estimate the optimal amount of data in tokens for a given compute budget (and model size). However, expressing data volume in tokens overlooks a critical aspect: the information density that each token represents. Consequently, scaling findings inherently depend on specific tokenizers and their key property: the compression rate.

To fill the research gap, we introduce scaling laws that are aware of the compression rate $T$, defined as the average number of bytes per token in a given dataset. For that purpose, we need to vary the compression rate

without changing the vocabulary size (and thus the number of parameters). Therefore, in our experiments we rely on Byte Latent Transformer (BLT, Pagnoni et al., 2025), a recent architecture that segments byte-level input in a latent space. BLT's latent tokenization is a robust tool for this purpose, as it allows us to precisely adjust the compression rate by setting an average segment size.[1] Additionally, compression plays a significant role in subword tokenization. We can order popular subword methods by their compression rate: from pure byte or character-level segmentation ($T \approx 1$) (Xue et al., 2022; Wang et al., 2024), through widely used Byte Pair Encoding (BPE) ($T \approx 4.57$) (Sennrich et al., 2016), to SuperBPE ($T \approx 6.16$) (Liu et al., 2025), which achieves high compression by allowing multi-word tokens.[2]

In the context of scaling, compression rate impacts model efficiency in both training and inference. Increasing compression allows the same data to be represented with fewer tokens, directly reducing the computational cost of processing. The unlocked savings in FLOPs (unit of computation) can be used to increase training data, model size, or both, without increasing the total computation budget.

To the best of our knowledge, this is the first thorough study of the effect of compression rate on the compute efficiency of language models. We pose the following research questions:

[**R1**] **How does compression rate impact the compute-optimal ratio between parameters and data?** This question concerns the unit of data we should use in model scaling. We investigate whether the compute-optimal ratio is best expressed in tokens or bytes (which are the underlying unit of text encoding). For example, given the Chinchilla rule of thumb of training on $\approx 20$ tokens per parameter (Hoffmann et al., 2022), does this ratio hold as we increase compression, or should the ratio of bytes to parameters remain constant given a dataset of English texts?

[**R2**] **Is there an optimal compression rate for specific datasets?** We investigate whether there exists a compression rate that yields the lowest loss for a fixed compute budget, assuming the optimal data to parameter ratio. Furthermore, we examine whether this optimal compression rate shifts with the compute budget, dataset, or task.

[**R3**] **Is the impact of compression rate on scaling trends similar for latent and subword tokenized models?** Does the answer to the previous questions depend on the tokenization method? We conduct experiments on subword-tokenized models to validate if the scaling trends match those observed for BLT.

[**R4**] **Is optimal compression rate language specific?** We extend our experiments to languages other than English to test whether optimal data to parameter ratio and compression rate change depending on language. We hypothesize that both will grow proportionally to *parity*, defined as the ratio of byte length of parallel sentences in different languages (Petrov et al., 2023; Ahia et al., 2023).

The structure of the paper is as follows. In Section 2, we describe background work and our experimental setting, including details of the datasets, models, and methods for deriving power laws. In Section 3, we present experiments scaling BLT across a wide range of compression rates to answer [**R1**] and [**R2**]. In Section 4, we examine subword-tokenized models to compare with the findings from the previous section and address [**R3**]. Finally, in Section 5, we extend our scaling experiments to languages other than English to answer [**R4**].

## 2 Methodology

In this section, we describe model architectures, training and evaluation procedures. We define variables used to describe the compute optimal scaling trends and outline methods for fitting power laws to model these trends.

### 2.1 Background: Subword Tokenization

Byte Pair Encoding (BPE) (Sennrich et al., 2016) is the dominant subword tokenization method in modern language models. BPE iteratively merges the most frequent pair of adjacent symbols in a corpus, starting from

---

[1]Recent latent tokenized models allow achieving a wide range of compression rate, similarly to BLT. However, in other approaches compression rate cannot be precisely controlled due to reliance on a segment boundary predictor (Hwang et al., 2025; Nawrot et al., 2023) or whitespace supervision (Neitemeier et al., 2025; Slagle, 2024; Videau et al., 2025).

[2]Estimates of compression rate are computed for DCLM corpus (Li et al., 2024) consisting of "plain English" texts.

a base alphabet of bytes or characters, until a target vocabulary size $V$ is reached. Each merge produces a new token representing a longer byte sequence. As the core BPE implementation, we use Llama3 (Llama Team, AI @ Meta, 2024) tokenizer with vocabulary of $V = 126{,}000$, yielding compression rate of approximately $T \approx 4.57$ bytes per token on English texts. A key property of BPE for our study is that its compression rate is fixed once the vocabulary is trained.

To vary the effective compression within the BPE framework, we employ **vocabulary masking**: we disable a fraction of the learned merge rules (masking 75% or 90% of the vocabulary), forcing the tokenizer to produce shorter, more frequent tokens. The tokenizers differ from base BPE mainly by achieving lower compression rates ($T = 4.16$ and $T = 3.71$, respectively) while segmentation algorithm and vocabulary construction are the same (except the masking). We also consider a **character-level** representation as a special case of subword tokenization, which obtains low compression rate ($T = 1.01$).

At the other extreme, **SuperBPE** (Liu et al., 2025) relaxes the standard assumption that tokens are restricted within word boundaries. By allowing merges to cross whitespace, SuperBPE produces multi-word tokens that encode text with fewer tokens than a standard BPE vocabulary of equivalent size. In our experiments, SuperBPE ($V = 200{,}000$) achieves $T \approx 6.16$, representing the highest compression rate among subword methods we consider.

## 2.2 Background: Latent Tokenization

In contrast to subword tokenizers that commit to a fixed segmentation, **latent tokenization** learns to group input units (bytes or characters) into variable-length segments within the model. We describe this method of tokenization as "latent", because latent tokens (or segments or patches) appear only in model's internal representation, while the input and output are byte-level.

Recent works have introduced hierarchical model architecture to implement latent tokenization (Nawrot et al., 2023; Hwang et al., 2025; Neitemeier et al., 2025). Such an architecture comprises three modules: (1) an encoder that aggregates byte-level representations into latent tokens; (2) a global module operating on these latent tokens (the Transformer model described above); and (3) a decoder that maps latent representations back to the byte level for next-byte prediction. In contrast to hierarchical architecture, throughout the paper we refer to language models with subword tokenization as "isotropic", i.e., models whose modules operate on sequences of the same granularity, unlike in hierarchical models.

**Byte Latent Tokenization (BLT)** In our experiments, we use Byte Latent Transformer (BLT) architecture (Pagnoni et al., 2025). BLT utilizes entropy spikes to segment byte sequences into latent tokens, allowing us to control the compression rate by adjusting the entropy threshold. Crucially for our purposes, BLT segments byte sequences based on the entropy of the next byte: high-entropy positions indicate segment boundaries, allocating more compute where data complexity is greater. By adjusting the entropy threshold, we can precisely control the average patch size, and hence the compression rate, without changing the model's parameter count. This makes BLT an ideal testbed for studying the effect compression rate on scaling behavior across wide range: ($T \in \{1, 2, 4, 6, 8, 12\}$). A key deviation from the original BLT implementation is the omission of hash embeddings for byte n-grams. We omit these because n-grams can span more bytes than the latent tokens themselves, potentially interfering with the target compression rate.

## 2.3 Scaling Variables Definitions

Scaling laws allow us to predict optimal training regimes of language models. Given a pre-defined computation budget, our task is to select specific model size in parameters and training data amount to minimize the loss of the model. Throughout this paper, we use the following notation for the key variables governing compute-optimal scaling:

- **Compute budget ($C$)** expressed by floating point operations (FLOPs) quantify how many operations are performed during model training.

- **Model size** ($N$) in parameters. Unless stated otherwise, we consider global module parameters, excluding encoder, decoder, and embedding layers.

- **Training data size** ($D$) is the number of tokens used for training a language model. In this paper we prefer to quantify data as number of bytes of text used for training ($B$), because this value is tokenizer independent.

- **Byte-per-parameter** ($\rho$) is defined as $\rho = B/N$. This quantity captures the data-to-model proportion and is crucial for scaling analysis.

- **Compression rate** ($T$), defined as the average number of bytes per token on reference corpus $T = B/D$.

- **Validation loss** ($L$) measured in bits-per-byte (BPB) to enable fair comparison across different tokenizers.

## 2.4 Model Architectures

For all experiments, we train Transformer models (Vaswani et al., 2017) of varying parameter sizes, adhering to Llama 3 architectural choices (Llama Team, AI @ Meta, 2024). We follow a standard scaling recipe: increasing models' width and depth in a 1:1 ratio, meaning the number of heads equals the number of layers. The latent dimension size is set to 128 times the number of heads, and the feed-forward network uses 4× upscaling.

The exact architecture specifications for all models used in our study are presented in Appendix A.

## 2.5 Training and Evaluation

We train models under compute budgets ($C$) expressed in FLOPs, ranging from $5 \times 10^{18}$ to $2 \times 10^{21}$ FLOPs. If not stated otherwise, we use exact computation of training FLOPs, instead of an approximation. In total, we train 988 latently and 320 subword tokenized models with sizes from 50M to 6.7B parameters on training data of sizes from 4B to 1.1T bytes.

For each budget $C$, we vary the parameter size ($N$) and compression rate ($T$). The parameters $N$ and compression rate $T$ uniquely determine the training data amount in bytes ($B$). Consequently, for each compute budget, we obtain a grid of models corresponding to the cartesian product of $T$ and $N$. For each of the configurations, we compute the bytes per parameter ratio ($\rho$), as shown in Figure 2. For BLT, we test six compression rate values $T \in \{1, 2, 4, 6, 8, 12\}$, while for subword models, compression rate is determined by the tokenizer $T \in \{1.01, 3.71, 4.16, 4.57, 6.16\}$. For all training runs, we fix the batch size at 2 million bytes and the learning rate at $4 \times 10^{-4}$. We use the AdamW optimizer (Loshchilov & Hutter, 2019) with a warmup-stable-decay learning rate schedule.

Unless stated otherwise, we train on DCLM (Li et al., 2024), a dataset of plain English texts selected to limit data mixing across domains and languages. Data mixing could cause non-uniform granularity of information and thus confound our analysis. We evaluate models on the C4 validation split (Raffel et al., 2020).

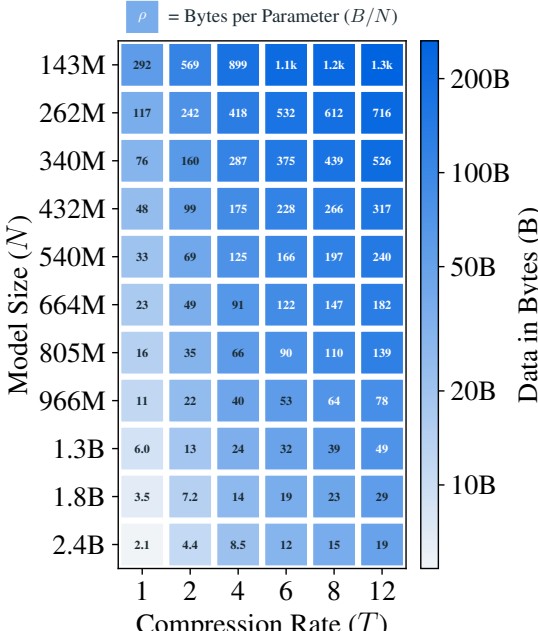

Figure 2: The grid of experiments for the budget of $C = 10^{20}$ FLOPs. Value of compression rate $T$ (x-axis) and model size $N$ (y-axis) determine the amount of training data $B$ (color) and the bytes per parameter ratio $\rho$ (values in squares).

To compare loss across different models with various tokenization methods, we evaluate models using bits-per-byte (BPB), which is loss divided by the number of bytes in the evaluation texts. In each training and

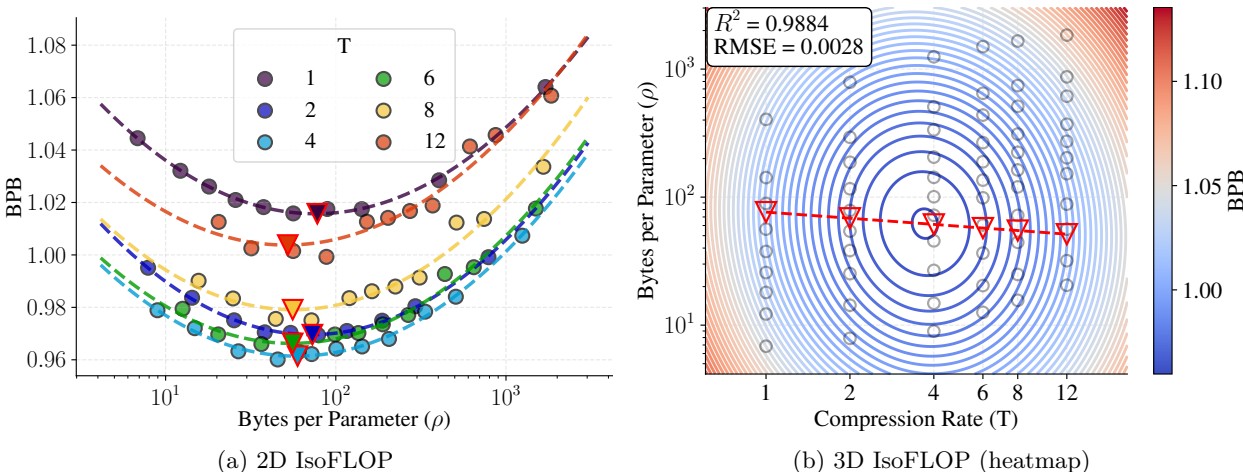

(a) 2D IsoFLOP

(b) 3D IsoFLOP (heatmap)

Figure 3: Evaluation scores of latent tokenized models on C4 test set with fixed FLOPs budget ($C = 10^{20}$), compared against bytes per parameter ratio. 2-dimensional IsoFLOP (parabola) were fitted for each compression rate, while 3-dimensional IsoFLOP jointly for all compression rates (on x-axis). Minima of both fits show that minimal loss is obtained at almost constant value of bytes per parameter ratio $\rho \approx 60$. For IsoFLOPs as function of data, parameters, and for other compute budgets, refer to Appendix F.1

evaluation example, we fix context to contain the same number of 8192 bytes (e.g., with compression rate $T = 4$ we evaluate on 2048 tokens per example, then with compression rate $T = 8$ we evaluate on 1024 tokens).

## 2.6 Fitting Power Laws

We fit the parameters for power laws presented in the next section using the BFGS optimizer (Liu & Nocedal, 1989; Zhu et al., 1997) minimizing sum of squares loss. To ensure reliability, we initiate optimization from multiple random seeds and compute confidence intervals using a numerical approximation of the Hessian. Further details on the fitting procedure can be found in Appendix B.

## 3 Scaling Laws and Data Compression

In this section, we present scaling results for BLT models revealing the role of data compression rate. We fit scaling laws in two stages, as such an approach shows more faithful approximations (Li et al., 2025). In the first stage, we estimate the optimal training data size in bytes $B^\star$ and model size $N^\star$ as a power law function of compute budget $C$ and compression rate $T$, addressing research question **R1**. Subsequently, in the second stage, we model the dynamics of the optimal loss $L^\star$ obtained for the found $B^\star$ and $N^\star$ configuration. We examine the effect of compression rate $T$ on $L^\star$ to answer research question **R2**.

### 3.1 Scaling Law I: Optimal Data and Parameters

For each compute budget $C$ and compression rate $T$, we identify the optimal training data size by fitting a second-degree polynomial (i.e. IsoFLOP) to the relationship between log-data $\log(B)$ and validation loss $L$. The optimal data size $B^\star$ corresponds to the minimum of this parabola. We determine the corresponding optimal parameter count $N^\star$ via log-linear interpolation.

In the first power law we estimate the optimal training data size $B^\star$ as a power law function of compute budget $C$ and compression rate $T$:

$$B^\star(C,T) \cong B_0 C^\alpha T^\beta \tag{1}$$

This equation involves three parameters: $B_0$ (initial optimal data), $\alpha$ (scaling with compute), and $\beta$ (scaling with compression). In this fit, for simplicity and better generalization across tokenizers, we consider only the parameters of the latent module (i.e., excluding encoder/decoder parameters for BLT and embedding parameters for subword models). Importantly, given our fixed scaling recipe, the number of the model's "latent" parameters determines the "total" parameter count. We can approximate the latent module's compute $C$ as:

$$C \approx 6N\frac{B}{T} \tag{2}$$

Where $\frac{B}{T}$ is the amount of data expressed in tokens, typically denoted as $D$ in other scaling laws works. Solving Approximation 1 allows us to obtain a power trend for optimal global parameter count:

$$N^\star(C,T) \cong \frac{1}{6B_0}C^{1-\alpha}T^{1-\beta} = N_0C^{1-\alpha}T^{1-\beta} \tag{3}$$

We also define the optimal Byte-per-Parameter ratio, $\rho^\star = B^\star/N^\star$. Based on the derived power laws, this ratio has the following form:

$$\rho^\star(C,T) \cong \frac{B_0}{N_0}C^{2\alpha-1}T^{2\beta-1} \tag{4}$$

Before observing the actual fit, we can describe the meaning of specific hypothetical values of $\alpha$ and $\beta$.

- When $\alpha \approx 0.5$, $\rho^\star$ would remain constant for varying values of compute budget $C$. This would mean that data and parameters should be scaled in 1:1 proportion. Similar equivalence was observed in Hoffmann et al. (2022).
- Analogously $\beta \approx 0.5$, would indicate that compute unlocked with higher compression should be allocated equally in increase of parameters and training data. Hence, the optimal bytes per parameter $\rho^\star$ would remain constant across varying compression rates $T$.
- $\beta \approx 1$ would indicate that we can omit the notion of compression from scaling laws and replace $B$ (amount of data in bytes) with used $D = \frac{B}{T}$ (amount of data in tokens). Such observation would suggest that we should simplify the scaling law to consider data amount in tokens $D^\star$ and neglect the impact of compression (as done in previous scaling studies).

### 3.2 Scaling Law I: Results

The IsoFLOPs analysis shows that for a set compute budget $C$, a second degree fit faithfully describes the relationship between logarithm of data size $\log(B)$ and validation loss $L$ (see Figure 18 in Appendix). Therefore, we can easily identify the optimal data size $B^\star$ by finding the minimum of the parabola (or paraboloid in the three-dimensional case).

Moreover, the results empirically confirm that the optimal data and parameter count gradually increase with increasing compression rate $T$, thanks to a decrease in compute cost per byte. Figure 3 indicates that across compression rates the optimal byte-per-parameter ratio $\rho^\star$ is close to constant. This implies that modifying tokenization (and thus compression rate) changes the compute optimal relation between tokens and parameters, whereas the relationship between bytes and parameters remains constant. Therefore, the latter is a more robust way to express the optimal data-to-model-size ratio, and we recommend considering it when designing language models with different tokenizers or vocabularies.

The fit reveals the following values of parameters: $B_0 = 11.1$, $N_0 = 1.5 \times 10^{-2}$, $\alpha = 0.475$, $\beta = 0.479$. Crucially, both the values of $\alpha$ and $\beta$ are close to 0.5, indicating that the optimal byte-per-parameter ratio is close to constant across varying compute budget and compression rate and 0.5 falls into 95% confidence intervals for the estimated value of $\beta$ (see Table 1). This allows us to answer the first research question **R1**:

> **Finding 1**
>
> The optimal ratio between bytes of data and model parameters ($\rho^\star$) remains close to constant across variable compute budget and compression rates. Therefore, when generalizing a scaling recipe to a model with a different tokenizer, we advise matching the ratio of training bytes (not tokens) to model parameters.

With the data-to-parameter ratio pinned down, a natural next step is to consider what is optimal value of compression rate that we should target at scale.

### 3.3 Scaling Law II: Optimal Loss Dynamics

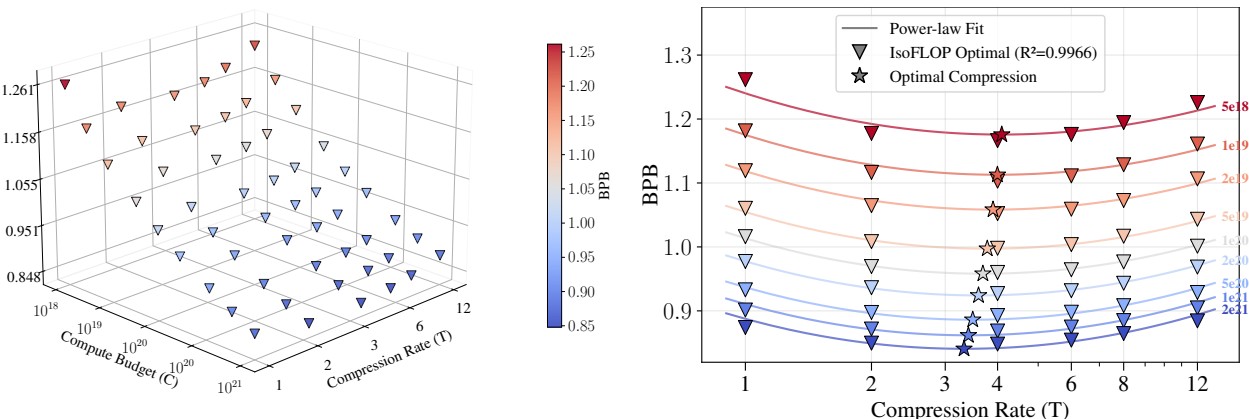

Figure 4: Optimal loss obtained for each compute budget and compression rate with latently tokenized models. Points at $C = 10^{20}$ correspond to the red triangles from Figure 3.

Figure 5: Power law fit for loss prediction based on compute budget and compression rate for BLT models. The slices of the fitted manifold for each compute budget (lines) are compared with the optimal loss values (triangles).

In the next stage, we model the optimal loss $L^\star$, defined as the loss obtained with the optimal data $B^\star$ and parameter count $N^\star$ for a given compute budget and compression rate:

$$L^\star(C,T) \stackrel{\text{def}}{=} L(B^\star(C,T), N^\star(C,T)) \tag{5}$$

We posit that the optimal loss can be approximated by a power law of the form:

$$L^\star(C,T) \cong L_0 \times C^\gamma + f(C,T) \tag{6}$$

This stage involves fitting three variables: $L_0$ (initial loss), $\gamma < 0$ (scaling with compute), and $f(T)$ (a function representing compression-specific residuals, including irreducible loss). We do not make a priori assumptions about the form of $f(C,T)$; instead, we fit it empirically based on the results obtained for each compression rate separately.

### 3.4 Scaling Law II: Results

We plot the optimal loss $L^\star$ as a function of compute budget $C$ and compression rate $T$ in Figure 4. While expectedly, the loss decreases with increasing compute budget, we observe that the relation between compression rate $T$ and $L^\star$ is non-monotonic. Specifically, the loss obtains a minimum for $T^\star \approx 4$ and rises for both higher and lower compression rates. We observe a slow decrease of optimal compression rate with increase of compute budget.

The power law fit gives us the following values of parameters: $L_0 = 3342$, $\gamma = -0.206$. We further examine the distribution of compression-specific offsets $f(C, T)$ in Figures 4 and 5. Based on the polynomial profile for $f(\cdot)$ we can estimate with high confidence its form as:

$$f(C, T) = F \times \log^2 \left( \frac{C^\delta T}{T_0} \right) + E, \tag{7}$$

where $F$ quantifies the contribution of compression rate on loss while $E$ is irreducible loss. The best fit was obtained with $F = 0.032$, $\delta = 0.035$, $T_0 = 18.2$, and $E = 0.70$. Both visual and power law evidence support the claim that the optimal compression rate $T^\star = \frac{T_0}{C^\delta}$ slowly decreases with training budget as evidenced by positive value of $\delta$ (with over 95% confidence). For instance, $T^\star = 3.69$ for $C = 10^{20}$ and $T^\star = 3.33$ for $C = 2 \times 10^{21}$. This allows us to answer the second research question **R2**:

> **Finding 2**
>
> At each training compute budget, there is an optimal compression rate $T^\star$. Diverging from its value in either direction increases loss. We observe decreasing optimal compression rate for higher training budgets.

The described scaling laws allow us to identify the values of bytes per parameter and compression rate for compute optimal training. Another practical consideration is how compute optimal compression rate impact computation cost of processing inference queries. We study it in the following subsection.

## 3.5 Optimal Tokenization during Inference

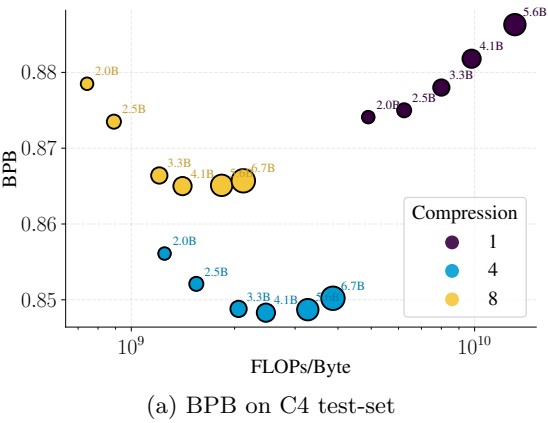

(a) BPB on C4 test-set

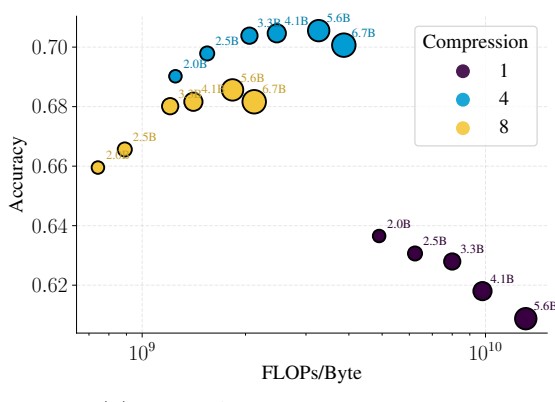

(b) 0-shot Accuracy on HellaSwag

Figure 6: Evaluation of the BLT models trained for $C = 2 \times 10^{21}$ FLOPs. Size of the point corresponds to model parameter count in the model. The results are plotted against inference compute cost per byte, which depends on model size $N$ and compression rate $T$.

To further study the role of optimal tokenization in inference, we compare the performance of models trained under $C = 2 \times 10^{21}$ budget with different compression rates against their inference cost. Specifically, we consider the results on language modeling and 0-shot accuracy on HellaSwag generative benchmark (Zellers et al., 2019). In Figure 6b, we observe that a higher compression rate decreases the inference compute cost for models of the same size (e.g. 3.3B parameter model with $T = 8$ is cheaper to run than a model of the same size $T = 4$). However, we also observe that compression rate closer to the optimal value improves the results of the inference-compute-matched setting. For instance, the 3.3B model with compression rate $T = 4$ has a similar inference cost of $2.1 \times 10^9$ FLOPs/Byte as the 6.7B model with compression rate $T = 8$, while the former achieves higher score on the endtask accuracy (74.1% vs. 68.2%). We present further results for AI2 Reasoning Challenge (Clark et al., 2018) in Appendix F.6.

Till now all the experiments were conducted for latent tokenized models. In the following section, we analyze whether the same trends hold for their counterparts with subword tokenization, e.g., BPE.

## 4 Compute Optimal Subword Tokenization

In this section, we validate the observations from the previous section for subword tokenized models. We train models with different subword tokenization algorithms: character-level tokenization, BPE, BPE with vocabulary masking, and SuperBPE to differentiate the values of compression rate $T$. Then we repeat the analysis of optimal data and parameters configurations and compare the fits of Scaling Laws I and II between latent and subword tokenized models.

### 4.1 Results

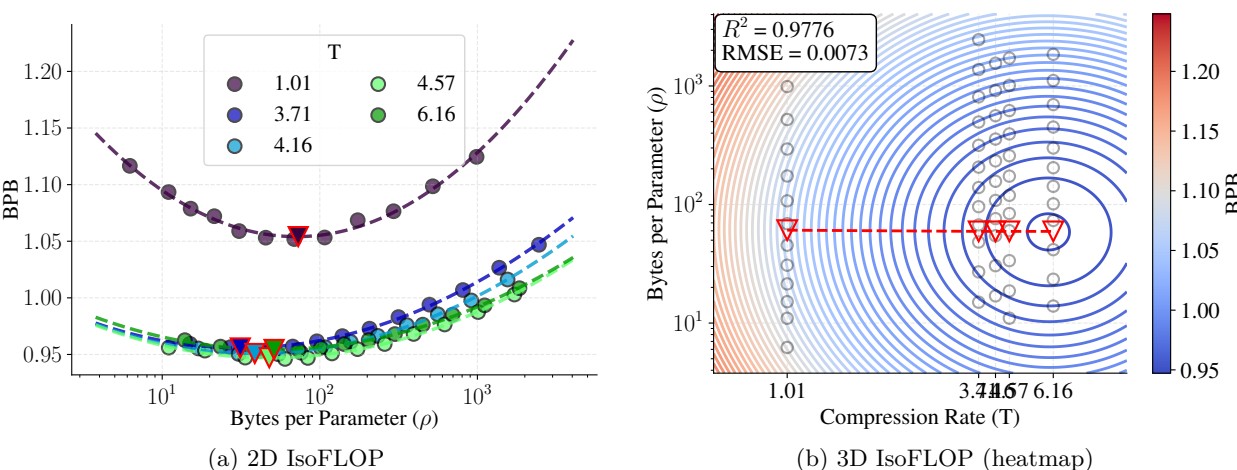

(a) 2D IsoFLOP  (b) 3D IsoFLOP (heatmap)

Figure 7: Evaluation scores of subword tokenized models on C4 test set with fixed FLOPs budget ($C = 10^{20}$), compared against bytes per parameter ratio. Different subword tokenization algorithms obtain varying compression rates: 1.01 for character-level tokenization, 3.71, 4.16, and 4.57 for BPE, 6.16 for SuperBPE 2-dimensional IsoFLOP (parabola) were fitted for each compression rate, while 3-dimensional IsoFLOP jointly for all compression rates (on x-axis). Similar to latent tokenized models, minima of both fits show that minimal loss is obtained for similar $\rho$ values across different compression rates. For IsoFLOPs as function of data, parameters, and for other compute budgets, refer to Appendix F.1

Similar to the latent tokenization case, the IsoFLOPs curves allow us to identify the optimal data amount in bytes $B^\star$ for a fixed compute budget $C$. Figure 7 shows the optimal values for a specific compute budget. We observe that the optimal byte-per-parameter ratio $\rho^\star$ is similar across tokenizers.

The Scaling Law I fit shows results close to the latent tokenization case: $B_0 = 2.8$, $N_0 = 59 \times 10^{-3}$, $\alpha = 0.501$, $\beta = 0.446$. Also, loss dynamics presented in Figure 8 and the Scaling Law II fit show similar results as the latent tokenization case: $L_0 = 1087$, $\gamma = -0.181$, $F = 0.0575$, $\delta = 0.129$, $T_0 = 1577$, and $E = 0.680$. The fit values are compared with latent tokenized models, as shown in Table 1.

Similar to the latent case, the Scaling Law II fit yields a positive compression–compute coupling $\delta = 0.129$, with a 95% confidence interval of $[0.108, 0.150]$ that

| Param | Latent | | Subword | |
|---|---|---|---|---|
| | Value | 95% CI | Value | 95% CI |
| $\alpha$ | 0.475 | [0.467,0.483] | 0.501 | [0.471,0.532] |
| $\beta$ | 0.479 | [0.458,0.500] | 0.446 | [0.387,0.506] |
| $B_0$ | 11.1 | [7.7,16.0] | 2.8 | [0.7,11.0] |
| $N_0$ | 0.015 | [0.0104,0.0217] | 0.059 | [0.015,0.229] |
| $\gamma$ | -0.206 | [-0.217,-0.195] | -0.181 | [-0.226,-0.135] |
| $L_0$ | 3342 | [2114,5286] | 1087 | [171,6896] |
| $F$ | 0.032 | [0.030,0.035] | 0.058 | [0.051,0.064] |
| $E$ | 0.703 | [0.688,0.718] | 0.680 | [0.618,0.741] |
| $\delta$ | 0.035 | [0.022,0.047] | 0.129 | [0.108,0.150] |
| $T_0$ | 18.2 | [10.2,32.4] | 1577 | [569,4369] |

Table 1: Fitted power law parameters for the families of latent and subword tokenized models. The 95% confidence intervals were computed with a numeric Hessian.

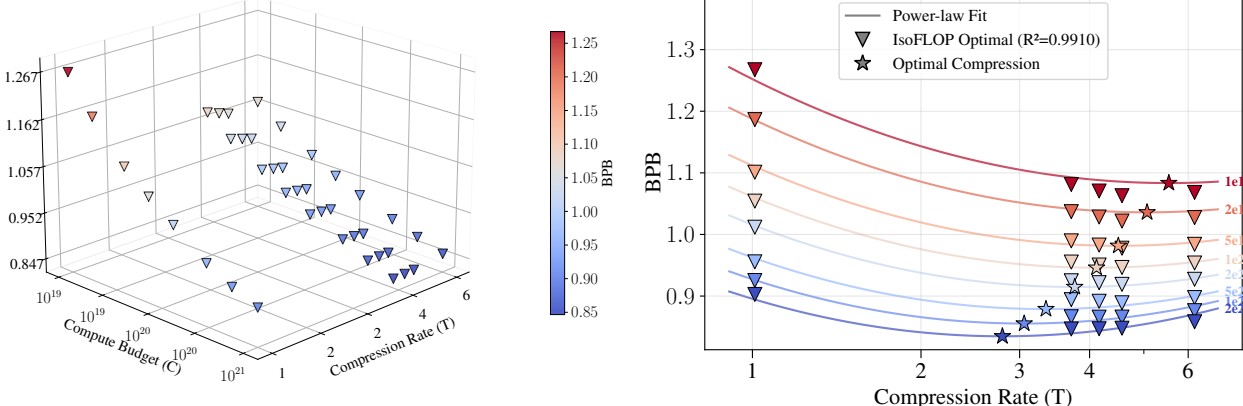

Figure 8: Optimal loss obtained for each compute budget and compression rate. Points at $C = 10^{20}$ correspond to data red triangles from Figure 7.

Figure 9: Power law fit for loss prediction based on compute budget and compression rate for isotropic models. The slices of the fitted manifold for each compute budget (lines) are compared with the optimal loss values (triangles).

excludes zero. This is our core evidence that the compute-optimal compression rate $T^\star$ decreases as the training budget grows, i.e. lower compression becomes preferable at larger scale (Figure 8). Consistent with this trend, at our highest compute budget the models with 75% and 90% of the BPE vocabulary masked reach a marginally lower BPB than the original BPE tokenizer, despite spending the same compute in the embedding and de-embedding layers (Table 2). We note that these differences are small and our claim that lower compression benefits larger-scale training rests on the statistically confirmed positive $\delta$ rather than on these pointwise comparisons.

The observations allow us to answer the question **R3**:

> **Finding 3**
>
> Discovered scaling trends for models with latent tokenization (BLT) hold for models with subword tokenization (BPE, SuperBPE).

We have thoroughly described the trends for compute optimal bytes per parameter and compression rate for training and evaluation on English datasets. In the next section, we are asking whether and how these findings generalize to other languages.

## 5 Compute Optimal Tokenization Beyond English

To test how the language choice affects the compute-optimal compression rate and bytes per parameter ratio, we extend our experiments to five languages with diverse writing scripts: French (Latin), Vietnamese (Latin), Russian (Cyrillic), Arabic (Arabic), Hindi (Devanagari). We also create an artificially inflated version of English data by adding a dummy byte between pairs of original UTF-8 bytes. Such English $\times 2$ data represent the same information at half the density.

For this purpose, we train latent-tokenized models (BLT) on monolingual data from FineWeb-2 (Penedo et al., 2025), training a separate set of models for each language. We evaluate each model on the corresponding test split from the same source. For English $\times 2$, we use an inflated version of the C4 test set used in the previous experiments.

| Compute | Character | BPE | | | SuperBPE |
|---|---|---|---|---|---|
| (FLOPs) | | V. mask=90% | V. mask=75% | Original | |
| $1 \times 10^{19}$ | 1.2678 | 1.0819 | 1.0709 | **1.0635** | 1.0682 |
| $2 \times 10^{19}$ | 1.1812 | 1.0381 | 1.0281 | **1.0214** | 1.0273 |
| $5 \times 10^{19}$ | 1.0989 | 0.9887 | 0.9819 | **0.9769** | 0.9840 |
| $1 \times 10^{20}$ | 1.0519 | 0.9554 | 0.9502 | **0.9461** | 0.9532 |
| $2 \times 10^{20}$ | 1.0126 | 0.9254 | 0.9220 | **0.9186** | 0.9272 |
| $5 \times 10^{20}$ | 0.9556 | 0.8942 | 0.8916 | **0.8891** | 0.8976 |
| $1 \times 10^{21}$ | 0.9253 | 0.8665 | **0.8658** | 0.8659 | 0.8763 |
| $2 \times 10^{21}$ | 0.9027 | **0.8466** | 0.8469 | 0.8479 | 0.8582 |
| **Compression:** | 1.01 | 3.71 | 4.16 | 4.57 | 6.16 |

Table 2: Comparison of the lowest BPB obtained by subword tokenized models for specific compute budgets.

Our training setup is analogous to the one described in Section 3 for English. For each language $l$, we fix the training budget to $C = 10^{20}$ FLOPs. The IsoFLOPs analysis (similar to that presented in Figure 3) allows us to identify the compute optimal bytes per parameter ratio $\rho_l^\star$ and compression rate $T_l^\star$, for each language $l$ at fixed compute budget.

Further, we compare these values to cross-lingual parity, defined as the proportion between the amount of bytes required to express the same information in different languages (Petrov et al., 2023). We estimate parity by dividing the byte length of sentences in each language by the byte length of their English translations. We use translations from FLORES-200 multi-parallel corpus (Goyal et al., 2021; team et al., 2022), which test split contains 1000 English sentences and their translations in a wide range of languages.

### 5.1 Results

Figure 10 present the results of the IsoFLOPs analysis across all analyzed languages. Similarly to English, we observe that the minimal loss is achieved by models with close to constant bytes per parameter ($\rho_l^\star$). From the polynomial fit, we estimate the compute-optimal bytes per parameter ratio and compression rate by analytically finding the coordinates of the global minimum (i.e. lowest loss).

| Language | Parity | $\rho_l^\star$ | | $T_l^\star$ | | BPB |
|---|---|---|---|---|---|---|
| | | Value | Ratio | Value | Ratio | |
| English | 1.0 | 62.1 | 1.0 | 3.71 | 1.0 | 0.960 |
| French | 1.2 | 57.8 | 0.93 | 4.16 | 1.12 | 0.795 |
| Vietnamese | 1.4 | 61.2 | 0.99 | 5.11 | 1.38 | 0.602 |
| Arabic | 1.6 | 75.8 | 1.22 | 4.58 | 1.23 | 0.564 |
| Russian | 2.0 | 96.3 | 1.55 | 5.67 | 1.52 | 0.461 |
| English (x2) | 2.0 | 91.1 | 1.47 | 6.97 | 1.87 | 0.492 |
| Hindi | 2.6 | 95.5 | 1.54 | 8.09 | 2.18 | 0.337 |

Table 3: Compute-optimal byte-per-parameter ($\rho_l^\star$), compression rate ($T_l^\star$) compared to cross-lingual parity. Results for monolingual models, with $C = 10^{20}$ FLOPs budget. The parity and compute-optimal ratios are proportions between each language and English baseline.

Notably, the $\rho_l^\star$ is language dependent (e.g. $\rho_{AR}^\star \approx 75.8$; $\rho_{RU}^\star \approx 96.3$). We also observe language-dependent differences in the compute-optimal compression rate (e.g. $T_{AR}^\star \approx 4.58$; $T_{RU}^\star \approx 5.67$). In Table 3, we compare these compute-optimal values to cross-lingual parity. We observe that the optimal values depend on language and its parity. Figure 11 shows that language-specific BPB scales inversely with parity. This confirms the observation from Limisiewicz et al. (2024): under optimal tokenization, similar information expressed across languages has similar likelihood. We further observe that parity correlates with optimal bytes per parameter, which is explained by the fact that more coarsely encoded languages tend to benefit more from additional training data than from larger models (Figure 12). While, in joint multilingual training the optimal bytes per parameter ratios converge to the same value across languages (see Appendix D).

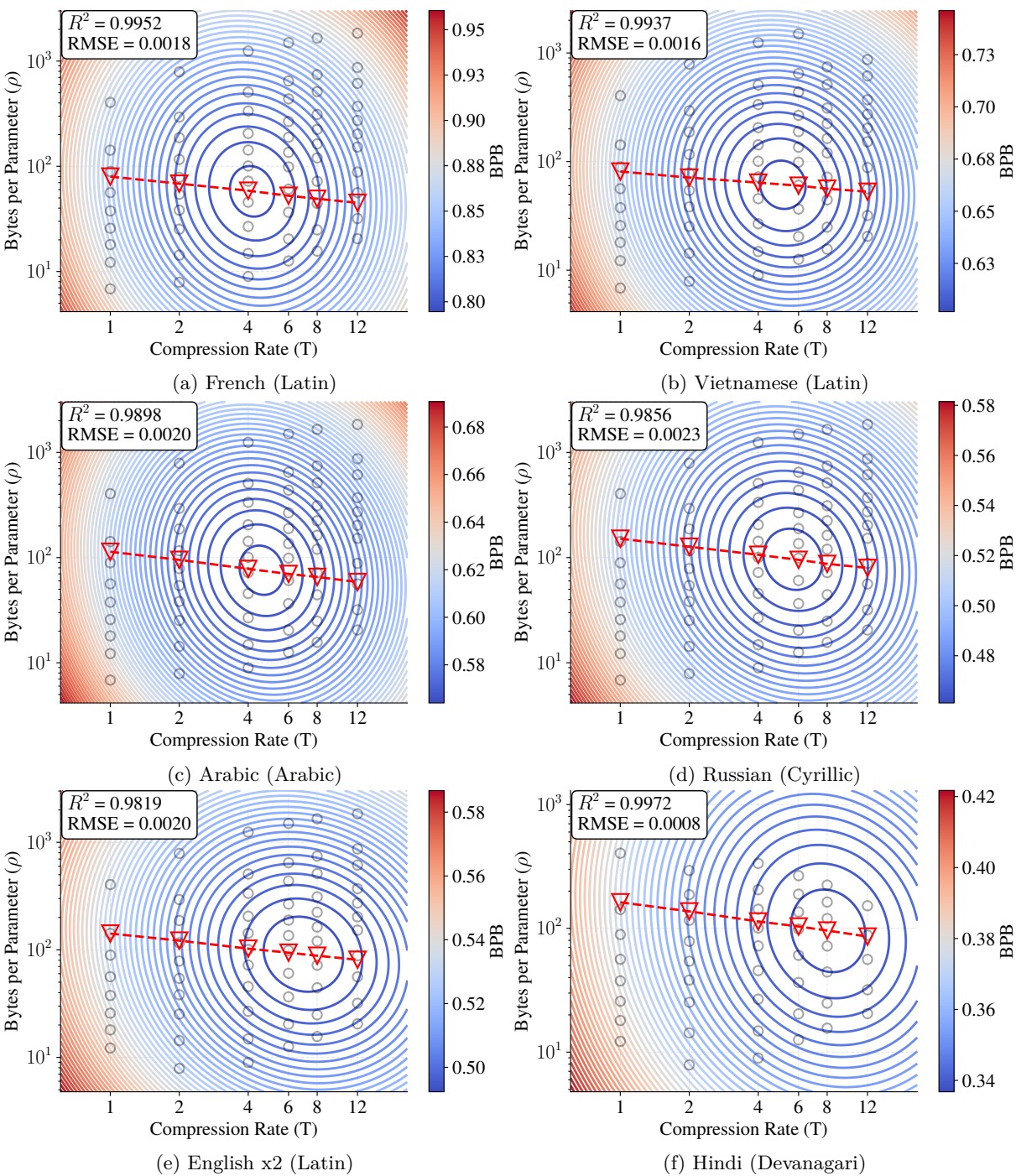

Figure 10: 3D IsoFLOP (heatmap) fits across languages ($C = 10^{20}$); all models use latent tokenization to achieve the set compression. Paraboloids are fitted jointly for all compression rates.

Lastly, we observe that higher parity translates to a higher optimal compression rate. For Latin-script languages, this relationship is close to a 1:1 increase (Figure 13). Importantly, the compression achieved by popular multilingual tokenizers: Llama 3 (Llama Team, AI @ Meta, 2024), Qwen 3 (Team, 2025), and EuroLLM (Martins et al., 2025), differs from the optimal value, as seen in Figure 14. These tokenizers tend to over-compress high-resource languages while under-compressing lower-resource ones.

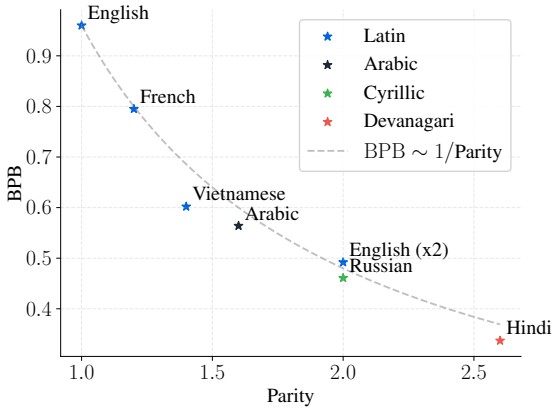

Figure 11: Language specific minimal loss compared against parity. We observe that BPB is inversely proportional to parity.

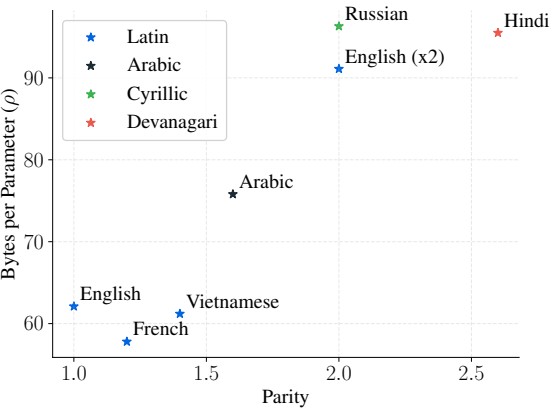

Figure 12: Language specific optimal bytes per parameter ratio ($\rho_l^\star$). Lower information density (high parity) correlates with preference of large training data size over model size (high $\rho_l^\star$).

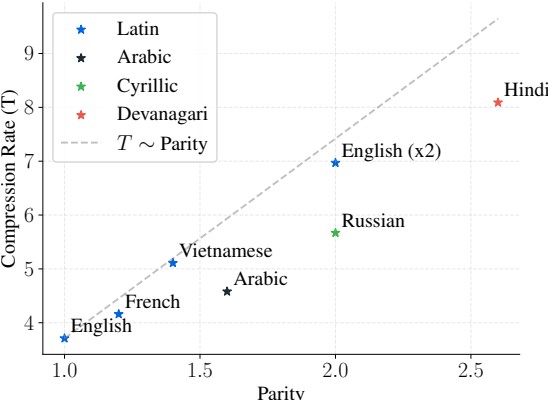

Figure 13: Language specific compression rate ($T_l^\star$) compared against parity. We observe that languages with higher parity prefer tokenization with higher compression.

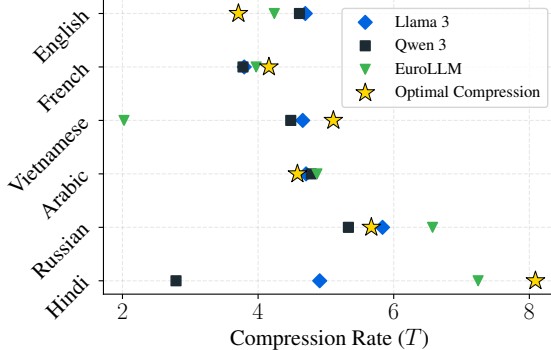

Figure 14: Compute optimal compression differs from the data compression obtained by popular language models. We observe that subword tokenizers over-compress high-resource languages (English, Arabic) while under-compressing the less resourced ones (Vietnamese, Hindi).

These results bring an answer to the last research question [**R4**]:

> **Finding 4**
>
> The optimal byte of data to parameter ratio ($\rho^\star$) and compression rate ($T^\star$) vary across different languages. Both are correlated with average information value of bytes in a given language (measured by parity).

## 6   Related Work

**Data Compression in Scaling Laws.**   Foundational scaling studies (Kaplan et al., 2020; Hoffmann et al., 2022) focus on model size and tokens, with subsequent works (Pearce & Song, 2024; Porian et al., 2024) noting that vocabulary embedding treatment causes divergence between them. Hoffmann et al. (2022) propose ≈20 tokens per parameter but assume fixed tokenization. We generalize this to $\rho^\star \approx 60$ bytes per parameter (for English). Tao et al. (2024) derived scaling laws for vocabulary size; we show benefits of larger vocabularies

diminish at scale.Recent works on domain- and language-specific scaling (Yang et al., 2025; He et al., 2025; Longpre et al., 2026) confirm that scaling behavior varies across settings, consistent with our multilingual findings.

**Search for Optimal Tokenization.** Higher compression rate has been linked to better performance in multilingual models (Rust et al., 2021; Limisiewicz et al., 2023; Goldman et al., 2024; Gallé, 2019), though compression may proxy for language frequency in training data. In English, Schmidt et al. (2024) argued higher compression is not always beneficial, while Liu et al. (2025) found downstream gains despite worse BPB. Our work reveals the relationship is non-monotonic: an optimal $T^\star$ exists beyond which performance degrades, and lower compression is preferred at larger scale. Prior assumptions favoring higher compression in multilingual settings may arise because subword tokenizers under-compress low-resource languages (Figure 1c).

**Scaling Latent Tokenized Models.** We use BLT (Pagnoni et al., 2025) for precise compression control. Recent latent tokenization results (Pagnoni et al., 2025; Hwang et al., 2025; Neitemeier et al., 2025; Nawrot et al., 2023) show hierarchical models surpass subword counterparts at sufficient scale. Latent tokenization also enables language-specific compression (Ahia et al., 2024; Owodunni et al., 2025), which our findings suggest is particularly beneficial for multilingual modeling.

# 7 Discussion

The relationship between scaling laws and data compression highlights the importance of considering tokenizer compression rate in the optimal design of large language models. Our observations overlap with the Hoffmann et al. (2022) (Chinchilla) recipe, suggesting that data and model parameters should be scaled proportionally. Generalizing the Chinchilla rule, we show that the appropriate unit for data quantity is bytes, not tokens. Therefore, the widely accepted rule of using approximately 20 tokens per parameter for compute-optimal training holds only under compression rate specific to a BPE tokenizer. We generalize this rule (Scaling Law I) by empirically showing that compute-optimal architectures for English text should use approximately 60 bytes per parameter, regardless of data compression. This generalization makes it easy to transfer efficient training settings across different tokenization schemes, spanning from byte-level to superword-level tokens.

Furthermore, Scaling Law II reveals the existence of an optimal compression rate that depends on the training domain and the compute budget. Interestingly, when training on English data with a small FLOP budget, the optimal compression rate is close to that of a BPE tokenizer. However, we observe a slow decrease as training compute increases. This observation also holds for subword-tokenized models. Strikingly, a model with 90% of the BPE vocabulary masked performs slightly better than standard BPE in our largest runs (even though both spend the same compute in the embedding and de-embedding layers). This surprising result suggests that, for compute-efficient training of large models, it could be beneficial to decrease vocabulary size or apply techniques such as BPE-dropout (Provilkov et al., 2020). Why do we observe such a counterintuitive result? Our hypothesis is that less-compressed tokenizers allow the model to use more compute at inference time by dividing each evaluation sample into more tokens that are processed by the model. It is important to keep in mind that lower compression naturally increases the cost of model usage (as shown in Section 3.5). Therefore, when controlling for compression rate, we should consider the trade-off between performance and inference cost. Specifically, it is advisable to use higher compression rate to decrease model usage cost, similarly to how model developers opt for over-training language models to boost the performance of relatively smaller (and thus cheaper) models compared to their training-compute-optimal counterparts.

The search for compute-optimal compression rate is especially important for languages other than English, where the compression obtained by subword tokenizers tends to diverge from the optimal value to a more extreme extent. Previously, it was thought that multilingual performance of language models is affected by over-segmentation, and many studies focused on increasing compression for better multilingual performance (Rust et al., 2021; Limisiewicz et al., 2023). We observe, across all considered languages, that overly high compression deteriorates results. Furthermore, for each language we find a specific optimal compression rate, the value of which is correlated with the relative information density of the text, i.e., *parity*. This observation highlights the importance of identifying and achieving an optimal compression rate for each of the modeled languages. For statistics-based subword tokenizers (such as BPE), compression rate is heavily impacted by

the amount of in-language data in the training corpus (Ahia et al., 2023) and encoding efficiency (Limisiewicz et al., 2024), and thus cannot be easily controlled in a massively multilingual setting. This limitation provides a strong argument for latent tokenizers in multilingual language modeling, whose compression can be adapted for specific languages (Ahia et al., 2024; Owodunni et al., 2025).

## 8   Conclusion

In this work, we have systematically studied the role of data compression on the scaling trend for large language models. We have shown that the optimal ratio between training data bytes and model parameters, denoted as $\rho^\star$, remains approximately constant across varying compute budgets and compression rates. Consequently, when generalizing scaling recipes to models with different tokenizers, we advise matching the ratio of training bytes (not tokens) to model parameters. Additionally, we find the optimal compression rate $T^\star$ that is specific to the training domain and slowly decreases with the training budget. Finally, we show that these scaling trends with compression rate hold consistently for both latent and subword-tokenized models.

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

## A  Model Scaling: Technical Details

In this section, we describe the model architectures in detail.

The core experiments were conducted with BLT models (Pagnoni et al., 2025). We followed the original implementation with a few notable exceptions. As noted in Section 2, we find that the local modules should be wide (high number of heads) and shallow (low number of layers). To achieve such a shape, we set the number of layers in each local module to the ceiling of one-fourth of the number of global layers, and the local head count to the ceiling of one-fourth of the number of global heads, plus 8. The cross-attention key-query duplication factor $k$ is set to the ceiling of one-eighth of the global module's head count. The hidden dimension of the local modules is set to 64 times the number of heads. This scaling recipe ensures that the compute overhead introduced by the local modules is comparable to the embedding layers found in isotropic models of similar scale.

An important divergence from the original BLT architecture of Pagnoni et al. (2025) is the omission of hash embeddings. To compensate for the reduced capacity for encoding the input, we increase the number of layers in the encoder to match the decoder (originally, the encoder has only one layer). Table 4 presents the scales and architecture hyperparameters of all BLT models used in this work.

Similarly, Table 5 outlines the scaling recipe for subword tokenized models.

We compare the compute spend in the latent module as a percentage of the total inference compute for both families of models in Table 6. We observe that with our scaling recipes, the global module takes up a similar share of compute in the BLT architecture as in the isotropic model when the model scale and compression rate are matched. We observe that decreasing compression rate or increasing model size correlates with a higher relative utilization of the global model.

## B  Scaling Laws: Technical Details

We characterize compute-optimal scaling through a two-stage fitting procedure.

### B.1  Scaling Law I

We fit the scaling laws to find optimal data and parameters as described in Equations 1 and 3. As noted in the methodology, we restrict this fit to the parameters of the global latent model (excluding encoder/decoder and embeddings) to ensure consistency across tokenization methods.

We perform the fit using the L-BFGS-B (Zhu et al., 1997) algorithm with a gradient tolerance of $10^{-10}$. To ensure robust convergence, we employ a grid search for initialization:

1. We first compute an Ordinary Least Squares (OLS) solution $(\alpha_{\mathrm{OLS}}, \beta_{\mathrm{OLS}}, B_{\mathrm{OLS}}, N_{\mathrm{OLS}})$ on the log-transformed data to serve as a prior.

2. We define a search grid by perturbing the OLS solution. We test 13 values for each parameter, resulting in $13^4$ total initialization points (though we fix $\alpha$ and $\beta$ ranges tighter than $B_0$ and $N_0$).

The grid is constructed as follows:

- $\log(B_{\mathrm{init}}) \in \{\log(B_{\mathrm{OLS}}) + \epsilon : \epsilon \in [-3, 3]\}$

- $\log(N_{\mathrm{init}}) \in \{\log(N_{\mathrm{OLS}}) + \epsilon : \epsilon \in [-3, 3]\}$

- $\alpha_{\mathrm{init}} \in \{\alpha_{\mathrm{OLS}} + \epsilon : \epsilon \in [-0.3, 0.3]\}$

- $\beta_{\mathrm{init}} \in \{\beta_{\mathrm{OLS}} + \epsilon : \epsilon \in [-0.3, 0.3]\}$

We select the solution that minimizes the sum of squares loss objective. The BFGS algorithm obtained the parameter values similar to OLS regardless of its starting point.

| Global (Latent Module) | | | | Local (Encoder/Decoder) | | | | Cross-Attention | | Total |
|---|---|---|---|---|---|---|---|---|---|---|
| Layers | Heads | Dim | Params | Layers | Heads | Dim | Params | Heads | k | Params |
| 5 | 5 | 640 | 25M | 2 | 10 | 640 | 10M | 10 | 1 | 50M |
| 6 | 6 | 768 | 43M | 2 | 10 | 640 | 10M | 10 | 1 | 68M |
| 7 | 7 | 896 | 67M | 2 | 10 | 640 | 10M | 10 | 1 | 93M |
| 8 | 8 | 1024 | 101M | 2 | 10 | 640 | 10M | 10 | 1 | 127M |
| 9 | 9 | 1152 | 143M | 3 | 12 | 768 | 21M | 12 | 2 | 199M |
| 10 | 10 | 1280 | 197M | 3 | 12 | 768 | 21M | 12 | 2 | 253M |
| 11 | 11 | 1408 | 262M | 3 | 12 | 768 | 21M | 12 | 2 | 318M |
| 12 | 12 | 1536 | 340M | 3 | 12 | 768 | 21M | 12 | 2 | 396M |
| 13 | 13 | 1664 | 432M | 4 | 12 | 768 | 28M | 12 | 2 | 506M |
| 14 | 14 | 1792 | 540M | 4 | 12 | 768 | 28M | 12 | 2 | 613M |
| 15 | 15 | 1920 | 664M | 4 | 12 | 768 | 28M | 12 | 2 | 738M |
| 16 | 16 | 2048 | 805M | 4 | 12 | 768 | 28M | 12 | 2 | 880M |
| 17 | 17 | 2176 | 966M | 5 | 14 | 896 | 48M | 14 | 3 | 1.1B |
| 18 | 18 | 2304 | 1.1B | 5 | 14 | 896 | 48M | 14 | 3 | 1.3B |
| 19 | 19 | 2432 | 1.3B | 5 | 14 | 896 | 48M | 14 | 3 | 1.5B |
| 20 | 20 | 2560 | 1.6B | 5 | 14 | 896 | 48M | 14 | 3 | 1.7B |
| 21 | 21 | 2688 | 1.8B | 6 | 14 | 896 | 58M | 14 | 3 | 2.0B |
| 22 | 22 | 2816 | 2.1B | 6 | 14 | 896 | 58M | 14 | 3 | 2.2B |
| 23 | 23 | 2944 | 2.4B | 6 | 14 | 896 | 58M | 14 | 3 | 2.5B |
| 24 | 24 | 3072 | 2.7B | 6 | 14 | 896 | 58M | 14 | 3 | 2.9B |
| 25 | 25 | 3200 | 3.1B | 7 | 16 | 1024 | 88M | 16 | 4 | 3.3B |
| 26 | 26 | 3328 | 3.5B | 7 | 16 | 1024 | 88M | 16 | 4 | 3.7B |
| 27 | 27 | 3456 | 3.9B | 7 | 16 | 1024 | 88M | 16 | 4 | 4.1B |
| 28 | 28 | 3584 | 4.3B | 7 | 16 | 1024 | 88M | 16 | 4 | 4.6B |
| 29 | 29 | 3712 | 4.8B | 8 | 16 | 1024 | 101M | 16 | 4 | 5.1B |
| 30 | 30 | 3840 | 5.3B | 8 | 16 | 1024 | 101M | 16 | 4 | 5.6B |
| 31 | 31 | 3968 | 5.9B | 8 | 16 | 1024 | 101M | 16 | 4 | 6.1B |
| 32 | 32 | 4096 | 6.4B | 8 | 16 | 1024 | 101M | 16 | 4 | 6.7B |

Table 4: The configuration of latent tokenized models (BLT architecture) used in scaling experiments.

| Global (Latent Module) | | | | Local Parameters (Embeddings) | | | Total Parameters | | |
|---|---|---|---|---|---|---|---|---|---|
| Layers | Heads | Dim | Params | Char | BPE | SuperBPE | Char | BPE | SuperBPE |
| 5 | 5 | 640 | 25M | 96M | 82M | 128M | 121M | 107M | 153M |
| 6 | 6 | 768 | 43M | 115M | 98M | 154M | 158M | 141M | 196M |
| 7 | 7 | 896 | 67M | 134M | 115M | 179M | 202M | 182M | 247M |
| 8 | 8 | 1024 | 101M | 154M | 131M | 205M | 254M | 232M | 306M |
| 9 | 9 | 1152 | 143M | 173M | 148M | 230M | 316M | 291M | 374M |
| 10 | 10 | 1280 | 197M | 192M | 164M | 256M | 389M | 360M | 453M |
| 11 | 11 | 1408 | 262M | 211M | 180M | 282M | 473M | 442M | 543M |
| 12 | 12 | 1536 | 340M | 230M | 197M | 307M | 570M | 536M | 647M |
| 13 | 13 | 1664 | 432M | 250M | 213M | 333M | 682M | 645M | 765M |
| 14 | 14 | 1792 | 540M | 269M | 229M | 358M | 808M | 769M | 898M |
| 15 | 15 | 1920 | 664M | 288M | 246M | 384M | 952M | 909M | 1.0B |
| 16 | 16 | 2048 | 805M | 307M | 262M | 410M | 1.1B | 1.1B | 1.2B |
| 17 | 17 | 2176 | 966M | 326M | 279M | 435M | 1.3B | 1.2B | 1.4B |
| 18 | 18 | 2304 | 1.1B | 346M | 295M | 461M | 1.5B | 1.4B | 1.6B |
| 19 | 19 | 2432 | 1.3B | 365M | 311M | 486M | 1.7B | 1.7B | 1.8B |
| 20 | 20 | 2560 | 1.6B | 384M | 328M | 512M | 2.0B | 1.9B | 2.1B |
| 21 | 21 | 2688 | 1.8B | 403M | 344M | 538M | 2.2B | 2.2B | 2.4B |
| 22 | 22 | 2816 | 2.1B | 422M | 360M | 563M | 2.5B | 2.5B | 2.7B |
| 23 | 23 | 2944 | 2.4B | 442M | 377M | 589M | 2.8B | 2.8B | 3.0B |
| 24 | 24 | 3072 | 2.7B | 461M | 393M | 614M | 3.2B | 3.1B | 3.3B |
| 25 | 25 | 3200 | 3.1B | 480M | 410M | 640M | 3.6B | 3.5B | 3.7B |
| 26 | 26 | 3328 | 3.5B | 499M | 426M | 666M | 4.0B | 3.9B | 4.1B |
| 27 | 27 | 3456 | 3.9B | 518M | 442M | 691M | 4.4B | 4.3B | 4.6B |
| 28 | 28 | 3584 | 4.3B | 538M | 459M | 717M | 4.9B | 4.8B | 5.0B |
| 29 | 29 | 3712 | 4.8B | 557M | 475M | 742M | 5.4B | 5.3B | 5.5B |
| 30 | 30 | 3840 | 5.3B | 576M | 492M | 768M | 5.9B | 5.8B | 6.1B |
| 31 | 31 | 3968 | 5.9B | 595M | 508M | 794M | 6.5B | 6.4B | 6.7B |
| 32 | 32 | 4096 | 6.4B | 614M | 524M | 819M | 7.1B | 7.0B | 7.3B |

Table 5: The configuration of subword tokenized models (isotropic). Parameter differences across tokenizers arise from varying vocabulary sizes $V$. For Character: $V = 148,000$, for BPE: $V = 128,000$, for SuperBPE $V = 200,000$.

| | Latent Tokenization | | | | | | Subword Tokenizetion | | |
|---|---|---|---|---|---|---|---|---|---|
| **Scale** | $T$=1 | $T$=2 | $T$=4 | $T$=6 | $T$=8 | $T$=12 | $T$=1.01 | $T$=4.57 | $T$=6.16 |
| 5 | 66% | 43% | 24% | 17% | 13% | 9% | 34% | 27% | 18% |
| 6 | 75% | 55% | 35% | 25% | 20% | 14% | 41% | 34% | 24% |
| 7 | 82% | 65% | 45% | 35% | 28% | 20% | 47% | 41% | 30% |
| 8 | 87% | 73% | 55% | 44% | 36% | 27% | 52% | 47% | 35% |
| 9 | 79% | 63% | 44% | 34% | 27% | 20% | 57% | 52% | 41% |
| 10 | 84% | 70% | 52% | 41% | 34% | 25% | 61% | 57% | 45% |
| 11 | 87% | 75% | 58% | 48% | 41% | 31% | 65% | 62% | 50% |
| 12 | 89% | 79% | 64% | 54% | 47% | 37% | 68% | 65% | 54% |
| 13 | 89% | 78% | 63% | 53% | 46% | 36% | 71% | 69% | 58% |
| 14 | 91% | 82% | 68% | 58% | 51% | 41% | 73% | 72% | 62% |
| 15 | 92% | 84% | 72% | 63% | 56% | 46% | 76% | 74% | 65% |
| 16 | 93% | 87% | 76% | 68% | 61% | 51% | 78% | 77% | 67% |
| 17 | 90% | 82% | 69% | 59% | 52% | 42% | 80% | 79% | 70% |
| 18 | 91% | 84% | 72% | 63% | 57% | 46% | 81% | 81% | 72% |
| 19 | 93% | 86% | 75% | 67% | 60% | 50% | 83% | 82% | 74% |
| 20 | 93% | 88% | 78% | 70% | 64% | 54% | 84% | 84% | 76% |
| 21 | 93% | 87% | 77% | 70% | 63% | 53% | 85% | 85% | 78% |
| 22 | 94% | 89% | 80% | 72% | 66% | 57% | 86% | 86% | 79% |
| 23 | 95% | 90% | 82% | 75% | 69% | 60% | 87% | 87% | 81% |
| 24 | 95% | 91% | 84% | 77% | 72% | 63% | 88% | 88% | 82% |
| 25 | 93% | 88% | 79% | 71% | 65% | 56% | 89% | 89% | 83% |
| 26 | 94% | 89% | 81% | 74% | 68% | 59% | 89% | 89% | 84% |
| 27 | 95% | 90% | 82% | 76% | 70% | 61% | 90% | 90% | 85% |
| 28 | 95% | 91% | 84% | 78% | 73% | 64% | 91% | 91% | 86% |
| 29 | 95% | 91% | 83% | 77% | 72% | 63% | 91% | 91% | 87% |
| 30 | 95% | 92% | 85% | 79% | 74% | 66% | 92% | 92% | 88% |
| 31 | 96% | 92% | 86% | 81% | 76% | 68% | 92% | 92% | 88% |
| 32 | 96% | 93% | 87% | 82% | 78% | 70% | 92% | 93% | 89% |

Table 6: The compute cost per byte by global model as percentage of compute cost per byte of the whole model. The first column (Scale) denotes number of layers and heads of global module. In latent tokenization compression rate $T \in \{1, 2, 4, 6, 8, 12\}$ is set as hyperparameter, whereas in subword tokenization it is determined by the tokenizer (Character $T = 1.01$, BPE $T = 4.57$, SuperBPE $T = 6.16$)

## B.2 Scaling Law II

In the second stage, we fit the power law for optimal loss $L^\star(C,T) \simeq L_0 C^\gamma + f(T)$. Unlike Stage I, we use the total compute budget, for this fit, including the cost of the encoder, decoder, and embeddings.

We fit the parameters $L_0$, $\gamma$, and the compression-specific offsets $f(T)$ simultaneously. We again use BFGS with a grid search for initialization. The grid spans 13 values for $L_0$ and $\gamma$ (169 combinations):

- $\log(L_{\text{init}}) \in [-3, 3]$

- $\gamma_{\text{init}} \in [-0.6, 0.0]$

The initial value for $f(T)$ is set to the mean loss observed at compression rate $T$. During optimization, we bound the parameters to physically plausible ranges: $\log(L_0) \in [-30, 30]$, $\gamma \in [-2, 0]$, and $f(T) \in [-5, 5]$.

## B.3 Derivation and Validation of Scaling Law II

| Residual model | $E$ | $F$ | $T_0$ | $\delta$ | RMSE | $R^2$ | $\bar{R}^2$ |
|---|---|---|---|---|---|---|---|
| Mean of residuals (Eq. 8) | 0.7075 | — | — | — | 0.0260 | 0.903 | 0.896 |
| Constant $T^\star$ (Eq. 9) | 0.7075 | 0.0341 | 3.73 | — | 0.0115 | 0.996 | 0.995 |
| Compute-dependent $T^\star$ (Eq. 7) | 0.7075 | 0.0341 | 14.9 | 0.0302 | **0.0086** | **0.997** | **0.996** |

Table 7: Comparison of the three considered forms for modeling $f(C,T)$ residuals in Equation 6. All functions were fitted using the 48 runs with compute budgets less than or equal to $1 \times 10^{21}$ FLOPs. To test extrapolation accuracy, Root Mean Square Error was computed for models trained at $2 \times 10^{21}$ FLOPs across 8 different compression rates. All evaluations of extrapolation performance and goodness-of-fit (standard and adjusted for the number of variables) indicate that the model with compute-dependent compression rate offers the best fit and extrapolation accuracy in loss estimation.

As described in Section 3.3, we begin the search for the scaling law equation by assuming the classical form from Kaplan et al. (2020), disregarding the role of compression. It is presented by Equation 6:

$$L^\star(C,T) \simeq L_0 \times C^\gamma + f(C,T)$$

First, we fit the first part of the scaling law, and then we examine the functions that would give the best approximation of $f(C,T)$ (residuals of the fit) with minimal complexity. We consider the following candidates for $f(C,T)$:

**Mean of the residuals** is equivalent to the "irreducible loss" term or intercept used in many scaling fits. It is the simplest form of $f(C,T)$, yet it still completely disregards the role of compression on loss. We consider the following form of irreducible loss:

$$f(C,T) = E \tag{8}$$

**Constant optimal compression** $(T^\star)$ is an assumption that the loss is always minimal for one compression rate, regardless of compute budget $C$. By an inspection of $f(C,T)$ residuals in Figure 5, we observe that they are distributed along a non-monotonic convex function of $T$, with a minimum at some point $T_0$. We assume that a quadratic function fits this relation well. Considering that $T$ is on a logarithmic scale, we propose the following equation for residuals (including also irreducible loss):

$$f(C,T) = F \times (\log(T) - \log(T_0))^2 + E = F \times \log^2\left(\frac{T}{T_0}\right) + E \tag{9}$$

**Compute-dependent optimal compression $(T^\star)$**   is based on a hypothesis that the optimal compression depends on compute budget. We observe that the minimum of the quadratic function modeling $f(T, C)$ described in the last paragraph shifts to a lower value with an increase of the training budget. To account for that, we include the effect of the compute budget in the log-quadratic function, arriving at the following formulation of Equation 7:

$$f(C, T) = F \times \log^2\left(\frac{C^\delta T}{T_0}\right) + E$$

To validate the extrapolation accuracy of the three candidate formulas, we fitted scaling laws for results of models trained for 8 computation budgets from $5 \times 10^{18}$ to $1 \times 10^{21}$, and 6 compression rates. For each compression rate and budget, we use the optimal model size (and training data) estimated by the Scaling Law I. Then we validate the obtained scaling laws by comparing expected vs. obtained loss for models trained with a higher compute budget: $2 \times 10^{21}$. In Table 7, we observe that the last formulation, making an assumption that the optimal compression rate is compute-dependent, obtains significantly lower mean square error in extrapolation than other candidate formulations. Moreover, the fit using this formula obtains the highest goodness-of-fit coefficient, both standard $(R^2)$ and adjusted for the number of fitted variables $(\bar{R}^2)$. Therefore, we decided to choose this formulation for the final version of the scaling law.

## B.4   Loss Sensitivity to Compression Rate

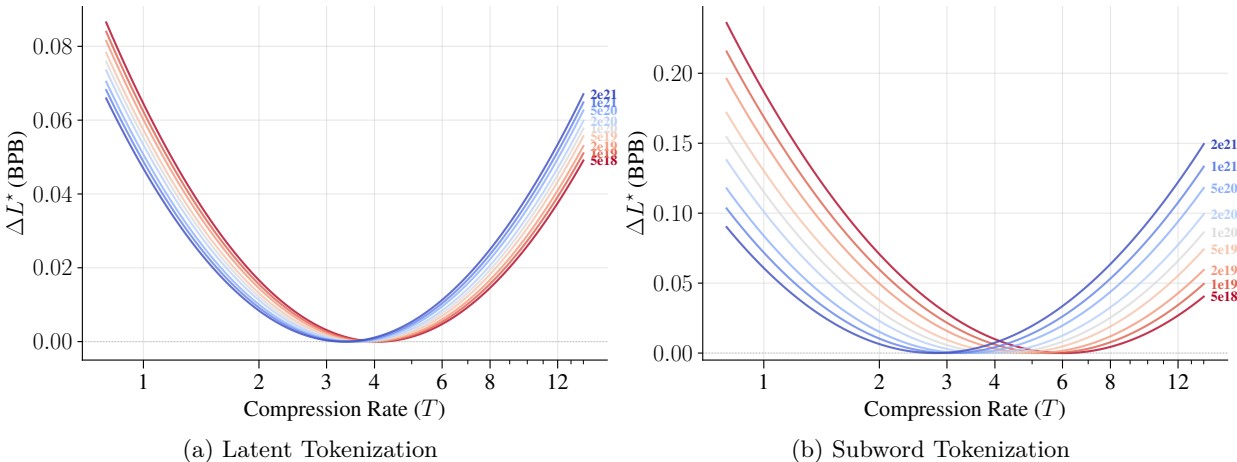

(a) Latent Tokenization

(b) Subword Tokenization

Figure 15: The BPB deterioration across compression compared to the value at optimal compression rate. $\Delta L^\star$ function was predicted based on Scaling Law II fit.

Figure 15 shows marginal sensitivity of loss to the choice of compression rate. We observe that compression rate close to optimal has minimal impact on loss, yet diverging further from the optimum can cause up to 0.2 and 0.1 deterioration in test BPB for subword and latent tokenized models respectively.

## B.5   Confidence Intervals

We compute 95% confidence intervals for the fitted parameters $\hat{\boldsymbol{\theta}} \in \mathbb{R}^p$ from $n$ data points, where $p$ is the number of parameters. $\mathcal{L}(\boldsymbol{\theta})$ denotes the sum of squares loss evaluated at $\boldsymbol{\theta}$, and $\mathbf{e}_k$ be the $k$-th standard basis vector in $\mathbb{R}^p$.

The Hessian $H \in \mathbb{R}^{p \times p}$ of $\mathcal{L}$ is estimated via central finite differences with step size $\epsilon = 10^{-5}$:

$$H_{ij} = \frac{\mathcal{L}_{ij}^{++} - \mathcal{L}_{ij}^{+-} - \mathcal{L}_{ij}^{-+} + \mathcal{L}_{ij}^{--}}{4\,\epsilon^2} \tag{10}$$

where

$$\mathcal{L}_{ij}^{s_1 s_2} = \mathcal{L}\big(\boldsymbol{\theta} + s_1\,\epsilon\,\mathbf{e}_i + s_2\,\epsilon\,\mathbf{e}_j\big) \qquad s_1, s_2 \in \{+, -\} \tag{11}$$

The residual variance is estimated as

$$\hat{\sigma}^2 = \frac{\sum_{i=1}^{n} r_i^2}{n - p} \tag{12}$$

where $r_i = y_i - \hat{y}_i$ is the $i$-th residual (observed minus predicted value). The parameter covariance matrix is

$$\hat{\Sigma} = \hat{\sigma}^2 \, H^{-1} \tag{13}$$

The 95% confidence interval for each parameter $\hat{\boldsymbol{\theta}}_k$ is:

$$\hat{\boldsymbol{\theta}}_k \pm t \cdot \sqrt{\hat{\Sigma}_{kk}} \tag{14}$$

where $\sqrt{\hat{\Sigma}_{kk}}$ is the standard error for estimation of $\hat{\boldsymbol{\theta}}_k$ and $t$ is the two-sided 95% critical value of Student's $t$-distribution with $n - p$ degrees of freedom.

## C Impact of Tokenization Method

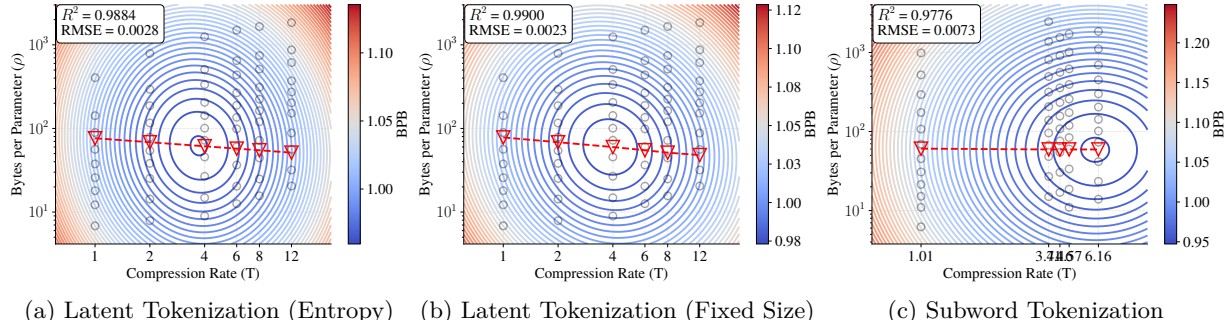

(a) Latent Tokenization (Entropy)    (b) Latent Tokenization (Fixed Size)    (c) Subword Tokenization

Figure 16: Comparison of three-dimensional IsoFLOPs ($C = 10^{20}$) for three methods of tokenization: Latent Entropy, Latent Fixed Size (each latent token has fixed size of $T$ bytes), and Subword. The loss profile is visibly similar across the methods, with optimal loss achieved along constant bytes per parameter.

| Tokenization Method | $\rho^\star$ | $T^\star$ | BPB |
|---|---|---|---|
| Latent (Entropy) | 62.1 | 3.71 | 0.960 |
| Latent (Fixed) | 60.0 | 3.87 | 0.973 |
| Subword | 58.8 | 5.36 | 0.947 |

Table 8: Compute-optimal bytes per parameter ($\rho^\star$) and compression rate ($T^\star$) for different methods. The values are close to each other, except for subword $T^\star$.

We compare our results for different methods of tokenization: latent (with entropy supervision) and subword, as described in the main text. Moreover, we compare the results with another method of latent tokenization, where all latent tokens are of the same fixed size in bytes equal to compression rate. In Figure 16, we see similar loss profiles across different methods. For all the methods and compression rates, the optimal configurations fall at $\approx 60$ bytes per parameter ratio ($\rho$). In Table 8, we further observe that for two latent tokenization methods the optimal compression rate is similar, while in subword tokenization it is higher. This is due to an imperfect IsoFLOP paraboloid fit caused by poor performance of character-level models ($T = 1.01$) under the considered budget, skewing optimal $T$ to be higher than in reality. Based on the more reliable Scaling Law II estimation (see Section 4) we expect to observe lower optimal compression rate for this budget: $T^\star = 4.11$. For comparison, optimal compression rate for latent models based on Scaling Law is $T^\star = 3.67$.

# D   Impact of Mixing Languages

| Language | Parity | $\rho_l^\star$ | | $T_l^\star$ | | BPB |
|---|---|---|---|---|---|---|
| | | Value | Ratio | Value | Ratio | |
| English | 1.0 | 71.8 | 1.00 | 3.38 | 1.00 | 1.101 |
| French | 1.2 | 72.5 | 1.01 | 3.65 | 1.08 | 0.931 |
| Vietnamese | 1.4 | 70.3 | 0.98 | 4.12 | 1.22 | 0.720 |
| Arabic | 1.6 | 76.5 | 1.07 | 3.84 | 1.14 | 0.667 |
| Russian | 2.0 | 77.6 | 1.08 | 5.03 | 1.49 | 0.532 |
| Hindi | 2.6 | 68.9 | 0.96 | 6.32 | 1.87 | 0.387 |

Table 9: Compute-optimal bytes per parameter ($\rho_l^\star$), compression rate ($T_l^\star$) compared to cross-lingual parity. Results for multilingual models, trained jointly on all six languages with $C = 10^{20}$ FLOPs budget. The parity and compute-optimal ratios are proportions between each language and English baseline.

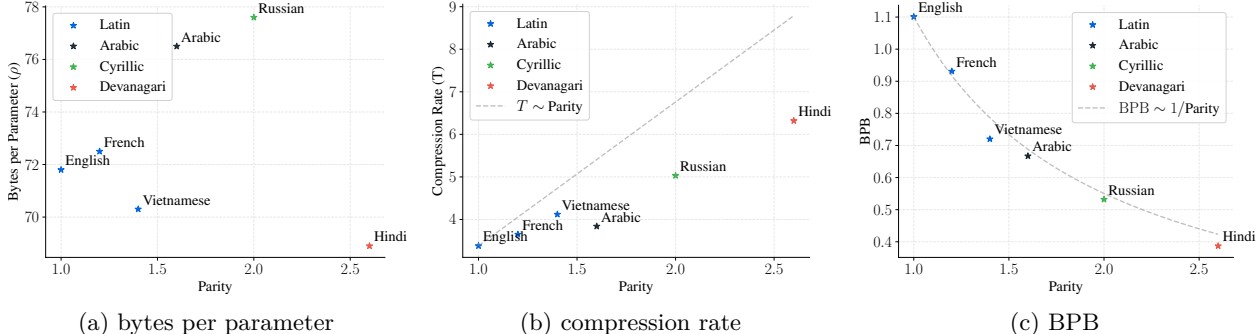

(a) bytes per parameter   (b) compression rate   (c) BPB

Figure 17: The optimal bytes per parameter, compression rate, and BPB for multilingual models trained on all languages jointly for $C = 10^{20}$ FLOPs. The optimal values are compared against parity on x-axis.

To examine the impact of mixing languages during training, we train a set of models jointly on multilingual data in six languages (including English), described in Section 5. To enforce an equitable training signal across languages, we sample languages with weights equal to their parity. For instance, we train on 2.6 more bytes in Hindi than in English, but we expect the two samples to be matched in information value. All training runs are constrained to a fixed budget of $C = 10^{20}$ FLOPs; thus, multilingual models see less in-language data per language than their monolingual counterparts.

The optimal values of bytes per parameter and compression rate for each language are computed based on fits to the in-language test set, the results are gathered in Table 9. Figure 17a shows that the optimal bytes per parameter is similar across languages. This contrasts with the findings in Section 5, where the optimal bytes per parameter was language-dependent and correlated with parity. Notably, the multilingual optimal bytes per parameter ($\rho^\star \approx 70$) is close to the median of the language-specific optimal values, $\rho_l^\star$. As in the monolingual experiments, we observe that the optimal compression rate (Figure 17b) is correlated with parity. The multilingual optimal values are lower than the corresponding monolingual ones. Test BPB (Figure 17c) is inversely correlated with parity, in line with the results of Section 5. As expected, multilingual models perform worse than monolingual ones due to the smaller amount of in-language data.

# E  Comparison with "Scaling Laws with Vocabulary"

Tao et al. (2024) posited similar research questions to ours regarding the role of tokenization in scaling laws, yet reached significantly different conclusions, showing that vocabularies (and thus compression rate) should increase with model scale. Meanwhile, we observe that the compute optimal compression rate does not increase with model scale. We identify the following methodological differences that explain discrepancies:

**Approach to embedding-layer compute and vocabulary size**   The main difference is how compression is connected to the size of the embedding layers. Tao et al. (2024) control compression rate by changing the vocabulary size, which affects the size of the embedding layer. This leads to a preference for smaller vocabularies at low compute and parameter budgets, so the FLOPs saved in embedding layers can be used for significantly longer training. In our experiments, vocabulary cost is (almost) the same regardless of compression, thanks to the use of BLT (Pagnoni et al., 2025) or alternative subword methods such as SuperBPE (Liu et al., 2025). Therefore, our results extrapolate better to larger scales, where the cost of the embedding layer is negligible, as seen in Table 6.

**Considered compression range**   BPE achieves a narrow compression rate range (by our estimates, $T \in [3, 4.5]$ bytes per token). Considering only compressions attainable by BPE allows us to observe only a portion of the loss profile, one that falls below the optimal compression value.

**Evaluation**   Both works use normalized negative log likelihood enabling a fair comparison across tokenizers. Tao et al. (2024) match validation context length in tokens, so the number of bytes in an evaluation example varies with vocabulary. We match the number of bytes across compression levels (e.g., if with compression rate $T = 4$ we evaluate on 2048 tokens per example, then with compression rate $T = 8$ we evaluate on 1024 tokens). Because early bytes are harder than later bytes, matching validation context in tokens can favor higher-compression tokenizers (more "late" bytes in an example). This could explain why we do not see the same preference for high compression (large vocabulary) at larger scales. For further reference, SuperBPE (Liu et al., 2025) also matched evaluation context in bytes. Similarly to our results, they observed worse BPB scores for highly compressed SuperBPE compared to regular BPE.

# F Supplementary Results

## F.1 IsoFLOP Analysis across Compute Budgets

We present the IsoFLOPs across multiple compute budgets and compression rates for latent tokenized models in Figures 18. And for subword tokenized models in Figures 20. We observe that the optimal byte-per-parameter ratio $\rho^\star$ remains constant for most of the considered configurations, this trend is more visible for $C > 10^{20}$, where the compute of the global module becomes dominant, thus it is expected to hold also at larger scales.

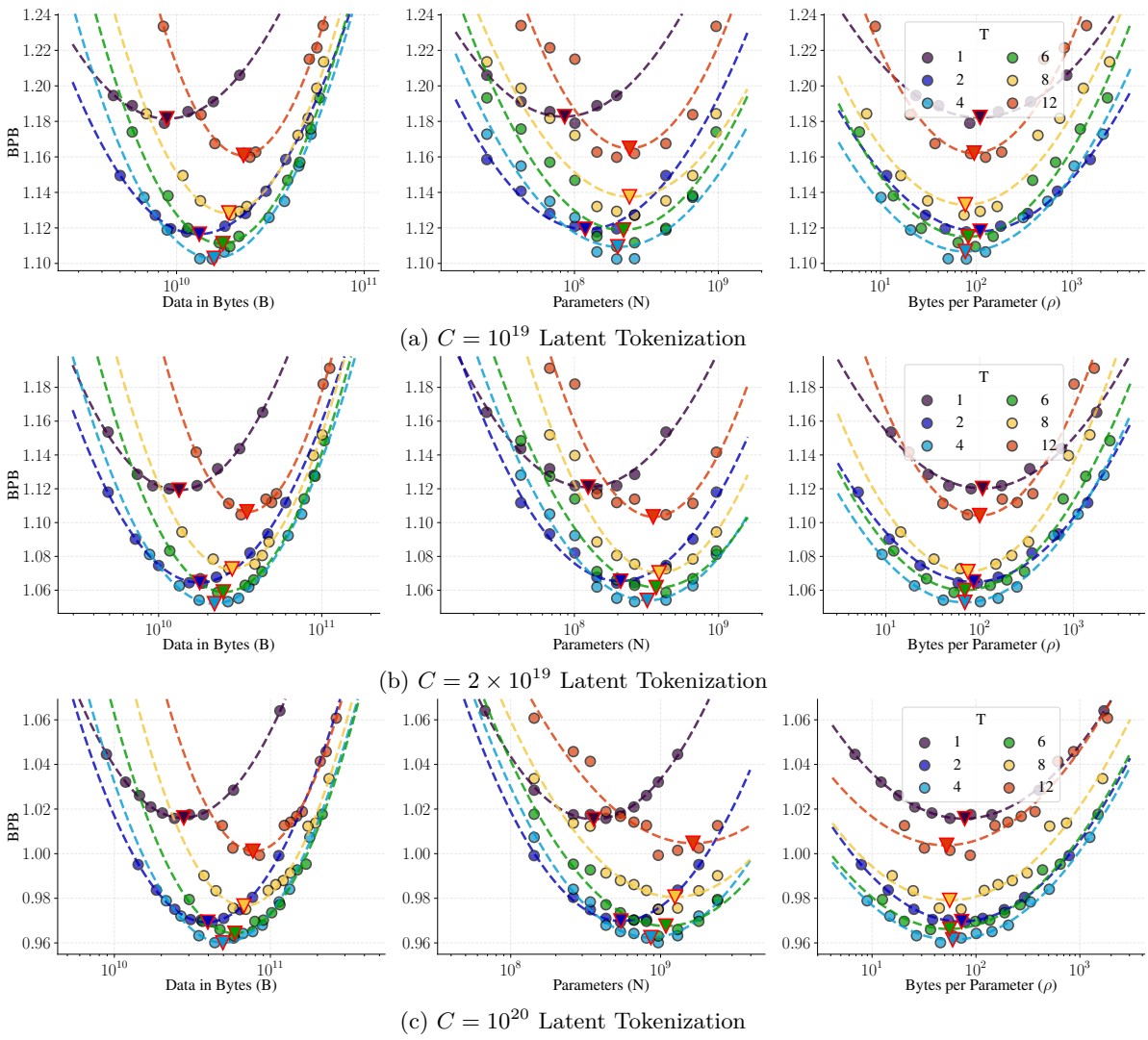

(a) $C = 10^{19}$ Latent Tokenization

(b) $C = 2 \times 10^{19}$ Latent Tokenization

(c) $C = 10^{20}$ Latent Tokenization

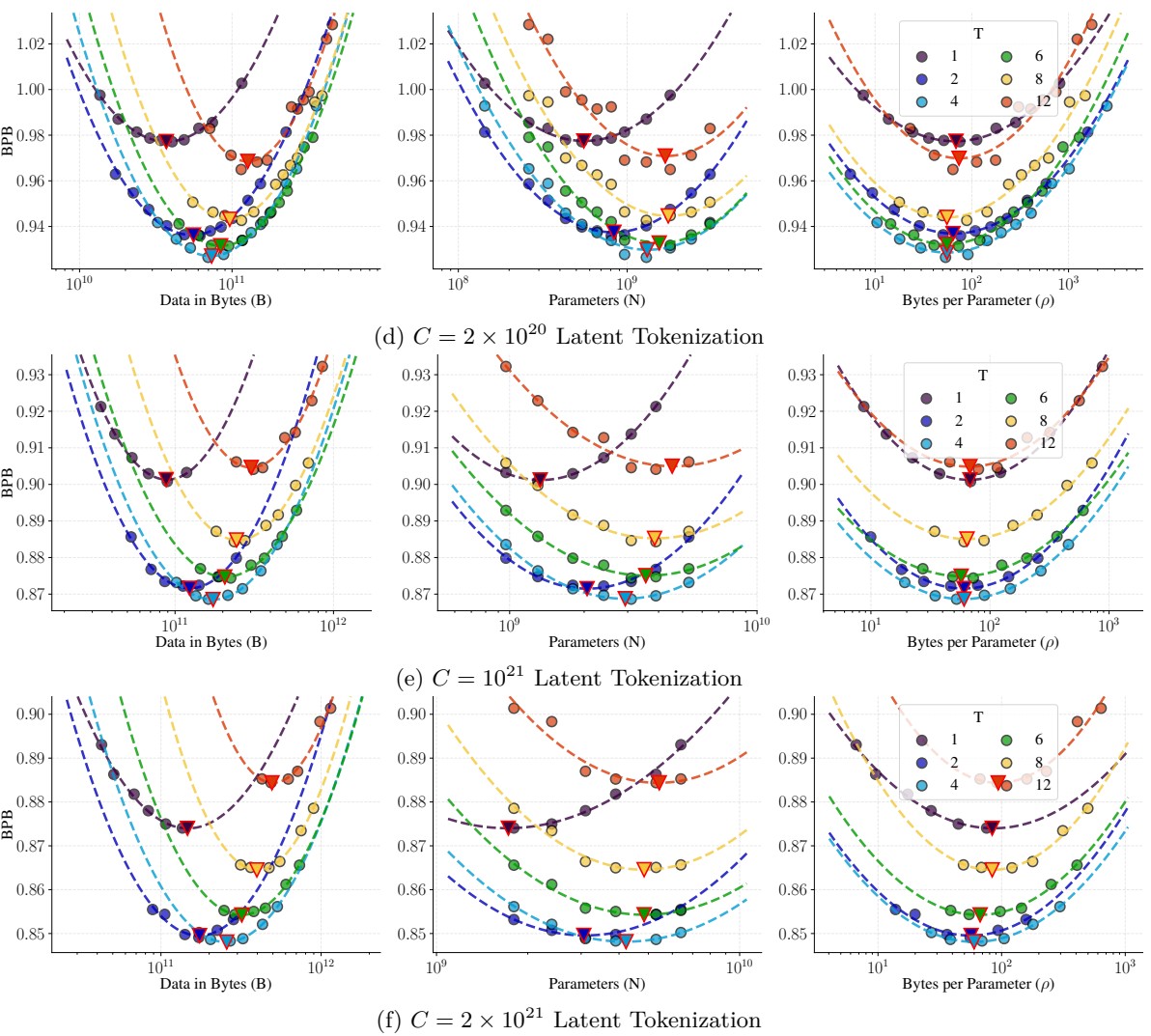

(d) $C = 2 \times 10^{20}$ Latent Tokenization

(e) $C = 10^{21}$ Latent Tokenization

(f) $C = 2 \times 10^{21}$ Latent Tokenization

Figure 18: 2-dimensional IsoFLOPs for latent tokenized models, as a function of data ($B$), parameters ($N$), or bytes per parameter ratio ($\rho$). Training budgets are indicated in each panel's caption. IsoFLOPs (parabolas) are fitted for each compression line to interpolate values of the loss.

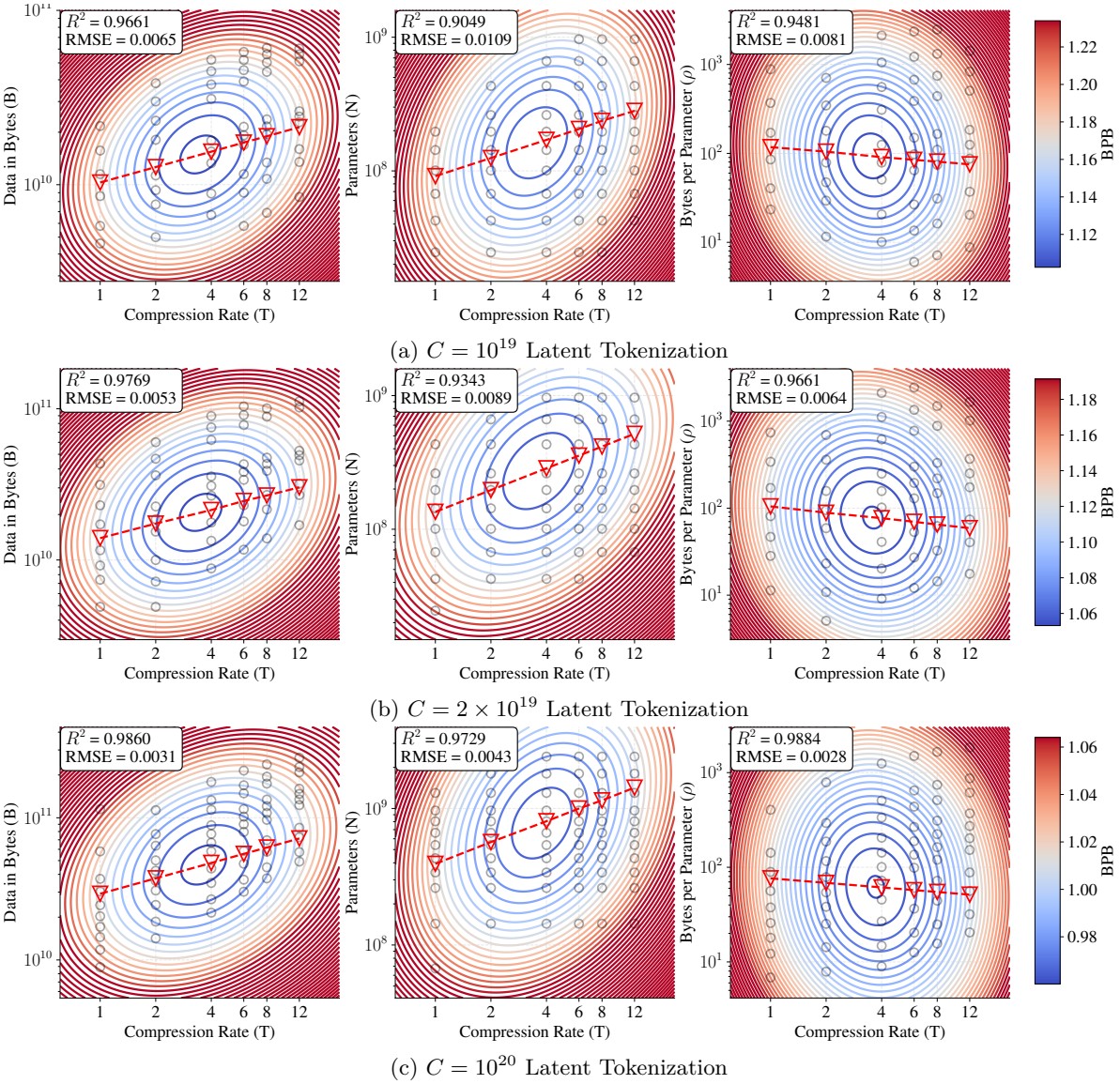

(a) $C = 10^{19}$ Latent Tokenization

(b) $C = 2 \times 10^{19}$ Latent Tokenization

(c) $C = 10^{20}$ Latent Tokenization

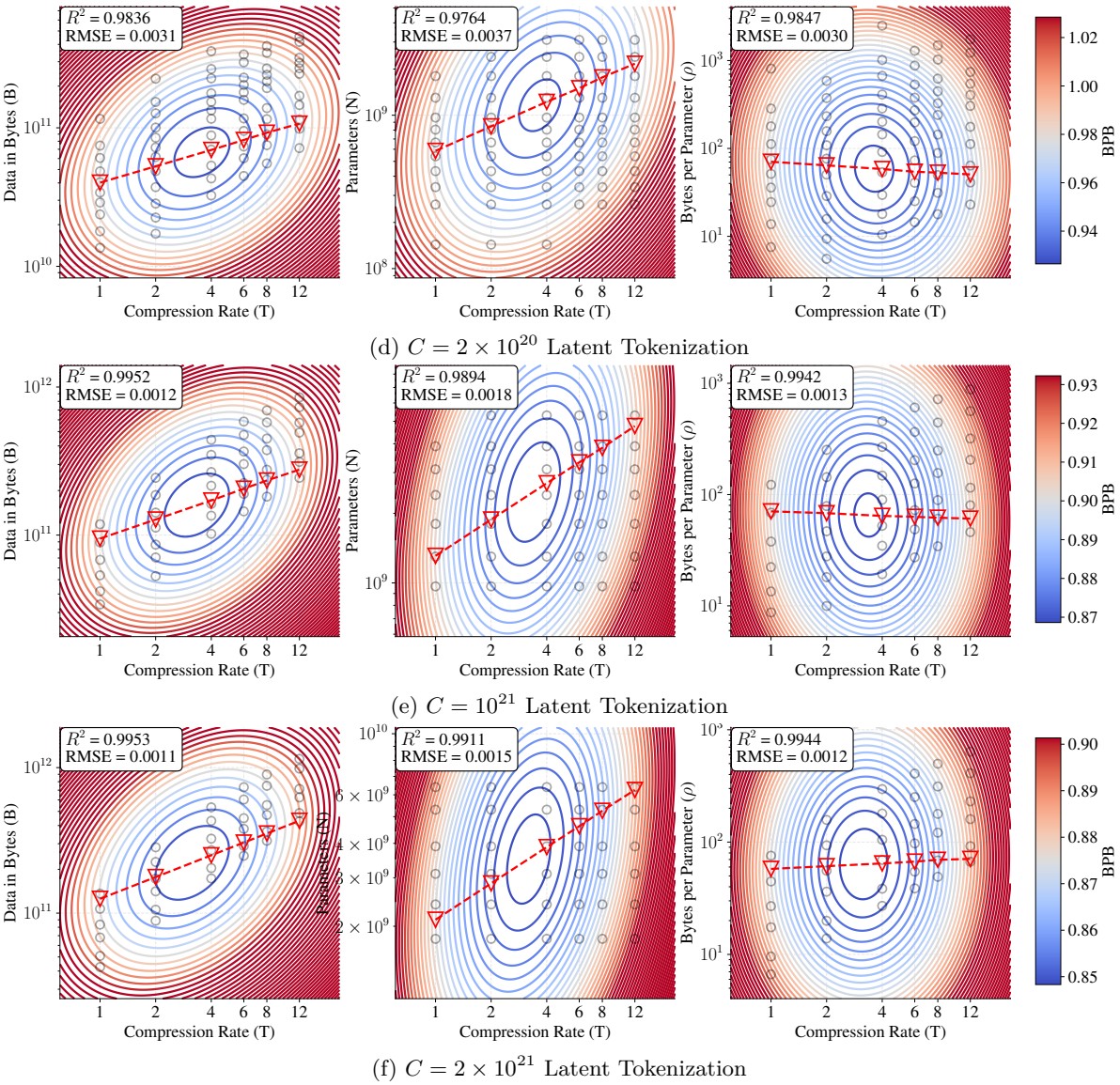

(d) $C = 2 \times 10^{20}$ Latent Tokenization

(e) $C = 10^{21}$ Latent Tokenization

(f) $C = 2 \times 10^{21}$ Latent Tokenization

Figure 19: 3-dimensional IsoFLOPs for latent tokenized models, as a function of compression rate and data ($B$), parameters ($N$), or bytes per parameter ratio ($\rho$). Training budgets are indicated in each figure's caption. IsoFLOPs (paraboloids) are jointly for all compression rates.

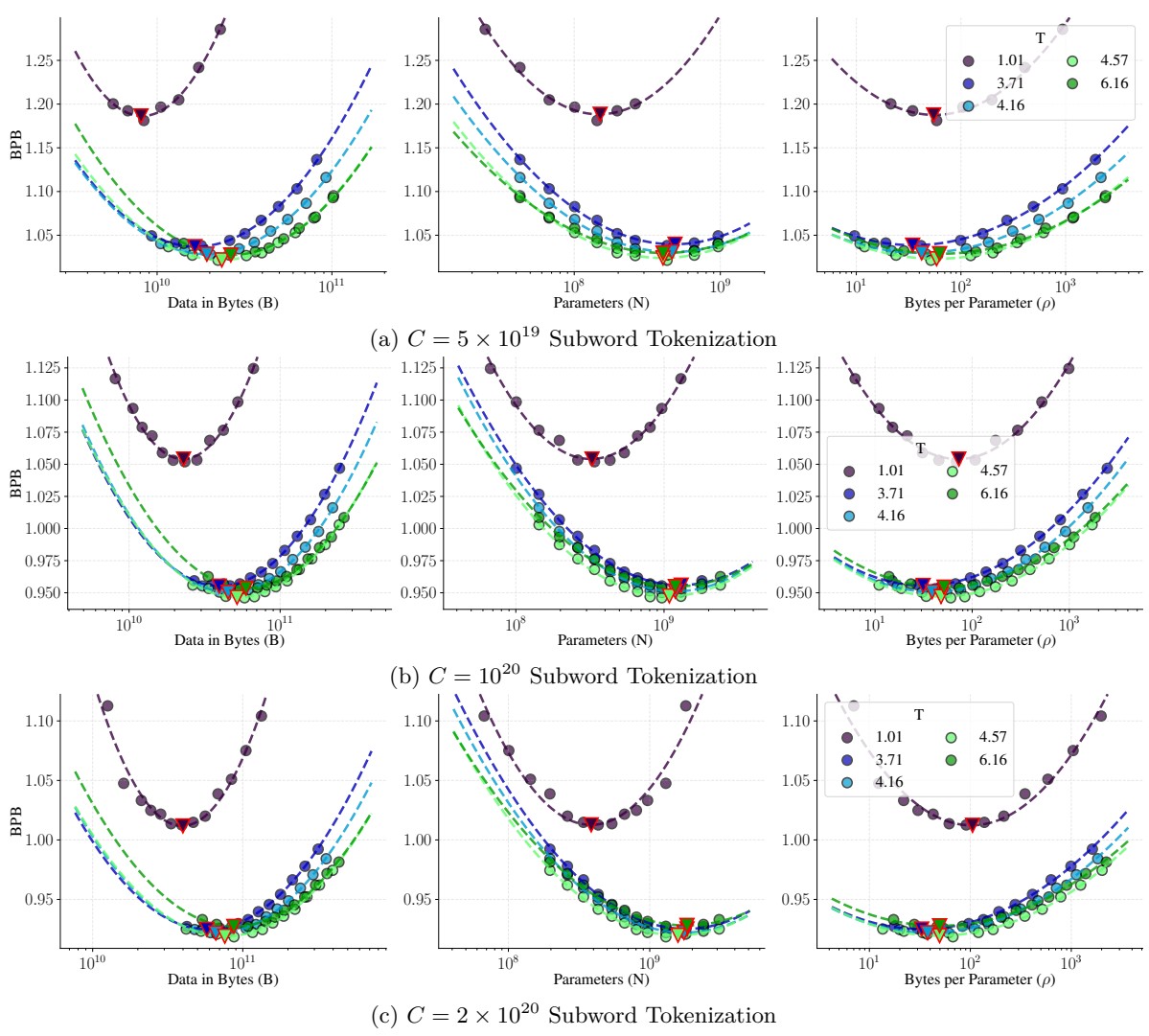

Figure 20: 2-dimensional IsoFLOPs for subword tokenized models, as a function of data ($B$), parameters ($N$), or bytes per parameter ratio ($\rho$). Training budgets are indicated in each panel's caption. IsoFLOPs (parabolas) are fitted for each compression line to interpolate values of the loss.

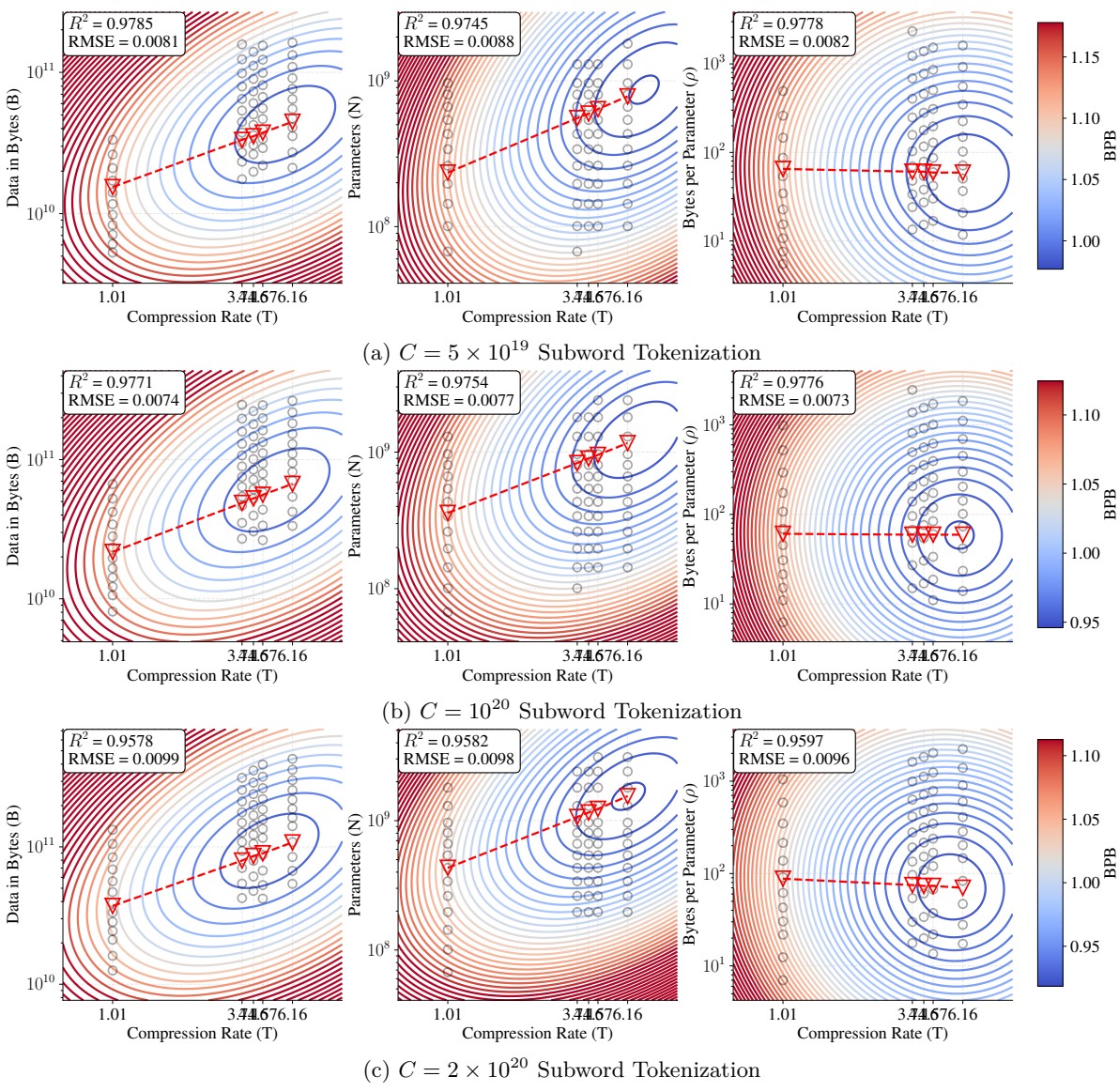

(a) $C = 5 \times 10^{19}$ Subword Tokenization

(b) $C = 10^{20}$ Subword Tokenization

(c) $C = 2 \times 10^{20}$ Subword Tokenization

Figure 21: 3-dimensional IsoFLOPs for subword tokenized models, as a function of compression rate and data ($B$), parameters ($N$), or bytes per parameter ratio ($\rho$). Training budgets are indicated in each figure's caption. IsoFLOPs (paraboloids) are jointly for all compression rates.

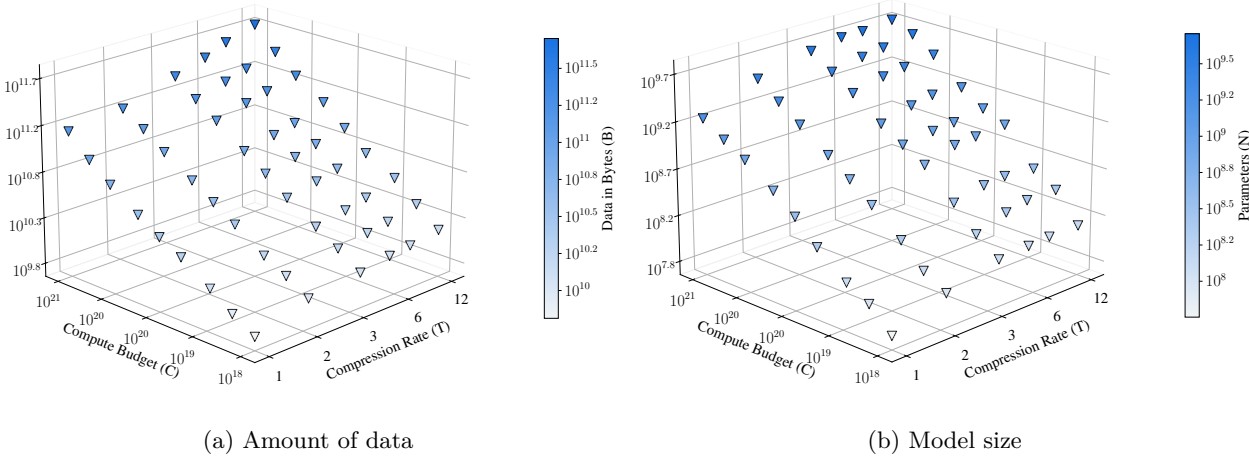

(a) Amount of data

(b) Model size

Figure 22: Optimal data and model size configurations for each compute budget and compression rate (latent tokenized models).

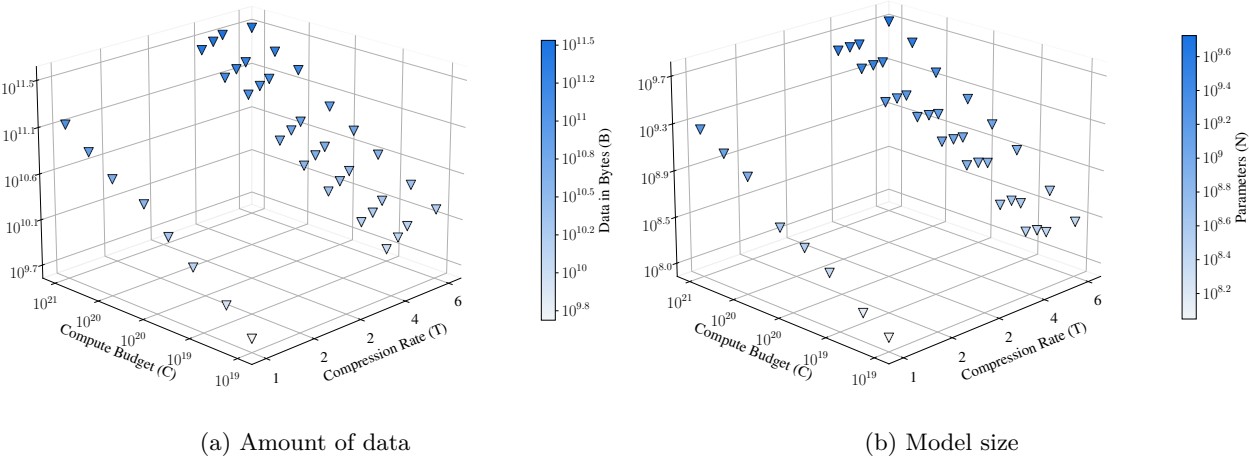

(a) Amount of data

(b) Model size

Figure 23: Optimal data and model size configurations for each compute budget and compression rate (subword tokenized models).

## F.2 Optimal Data and Parameters across Compute Budgets

Figure 22 shows the optimal data in bytes $B^*$ and parameter counts $N^*$ across compressions and compute budgets for latent tokenized models. Analogously, Figure 23 presents same data for subword tokenizers models.

## F.3 Loss Obtained for Optimal Configurations

In Tables 10 and 2, we present the best scores obtained by models (i.e., not derived from scaling law) respectively for latent and subword tokenized models.

| Compute | Latent Entropy | | | | | |
|---|---|---|---|---|---|---|
| (FLOPs) | 1 | 2 | 4 | 6 | 8 | 12 |
| $1 \times 10^{19}$ | 1.1790 | 1.1178 | **1.1025** | 1.1095 | 1.1272 | 1.1598 |
| $2 \times 10^{19}$ | 1.1200 | 1.0642 | **1.0532** | 1.0587 | 1.0727 | 1.1047 |
| $5 \times 10^{19}$ | 1.0606 | 1.0080 | **0.9987** | 1.0049 | 1.0165 | 1.0422 |
| $1 \times 10^{20}$ | 1.0158 | 0.9694 | **0.9601** | 0.9631 | 0.9751 | 0.9993 |
| $2 \times 10^{20}$ | 0.9771 | 0.9359 | **0.9265** | 0.9314 | 0.9427 | 0.9650 |
| $5 \times 10^{20}$ | 0.9333 | 0.8974 | **0.8933** | 0.8990 | 0.9085 | 0.9278 |
| $1 \times 10^{21}$ | 0.9008 | 0.8722 | **0.8686** | 0.8744 | 0.8843 | 0.9041 |
| $2 \times 10^{21}$ | 0.8741 | 0.8491 | **0.8483** | 0.8543 | 0.8650 | 0.8844 |
| **Compression:** | 1 | 2 | 4 | 6 | 8 | 12 |

Table 10: Comparison of the lowest BPB obtained by latent tokenized models for specific compute budgets.

### F.4 Multilingual IsoFLOP

In Figure 24, we present 2-dimensional IsoFLOP for six considered languages. The visualization is based on the same data as used for 3-dimensional IsoFLOP in Figure 10.

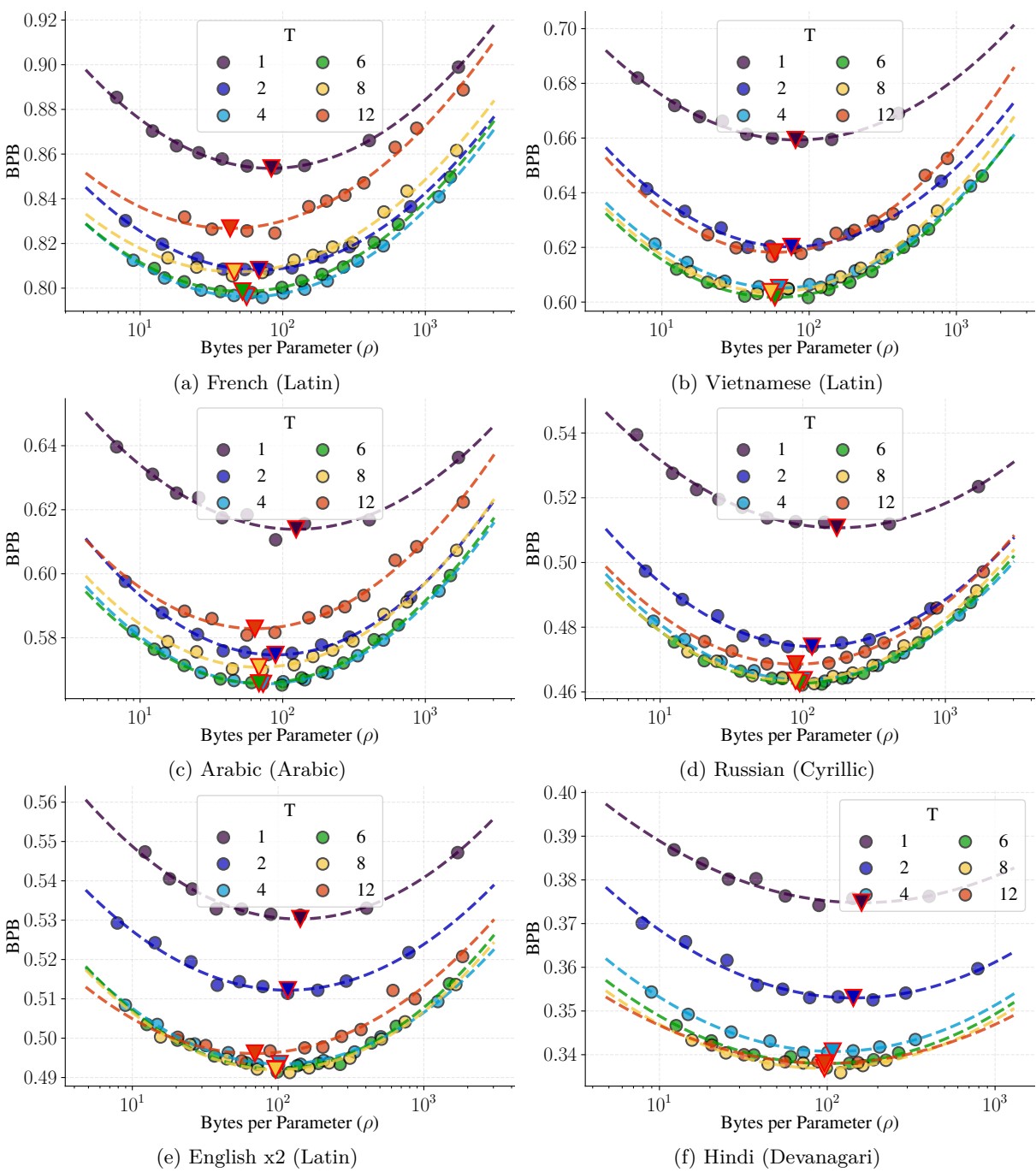

Figure 24: 2D IsoFLOP fits across languages ($C = 10^{20}$); all models use latent tokenization to achieve the set compression. Parabolas are fitted for each compression line to interpolate values of the loss.

### F.5 Comparison between Character and Byte-level Models

In our analysis of subword tokenized models we focus on character-based instead of byte-based models to examine the properties of low compression. The main difference between these models is that the former has a much larger vocabulary (148,000 vs. 256), while achieving a similar compression rate. In our experiments, we consider character models to coerce on a similar vocabulary size as in BPE and SuperBPE.

We compare the loss of parameter optimal character ($T = 1.01$) and byte models ($T = 1.0$) in Figure 25. Notably, the gap between them is large for a small compute budget due to the relatively high cost of the embedding layer in small models. With the increase of the training budget, the difference narrows. This allows us to assume that character and byte tokenized models will follow similar scaling trends at larger scales. Therefore, in the most of experiments we only consider character-based models.

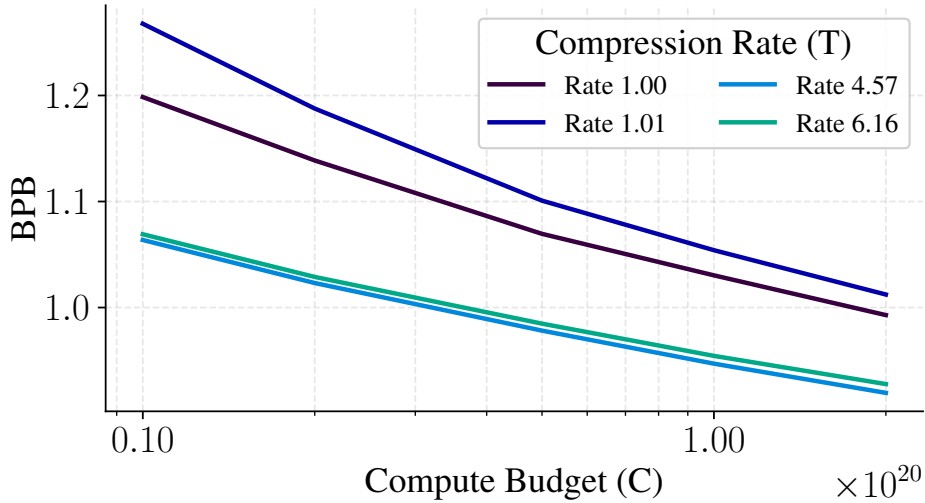

Figure 25: Comparison of optimal test losses for subword tokenized models: byte $T = 1.00$; character $T = 1.01$; BPE $T = 4.57$; SuperBPE $T = 6.16$.

## F.6 AI2 Reasoning Challenge Results

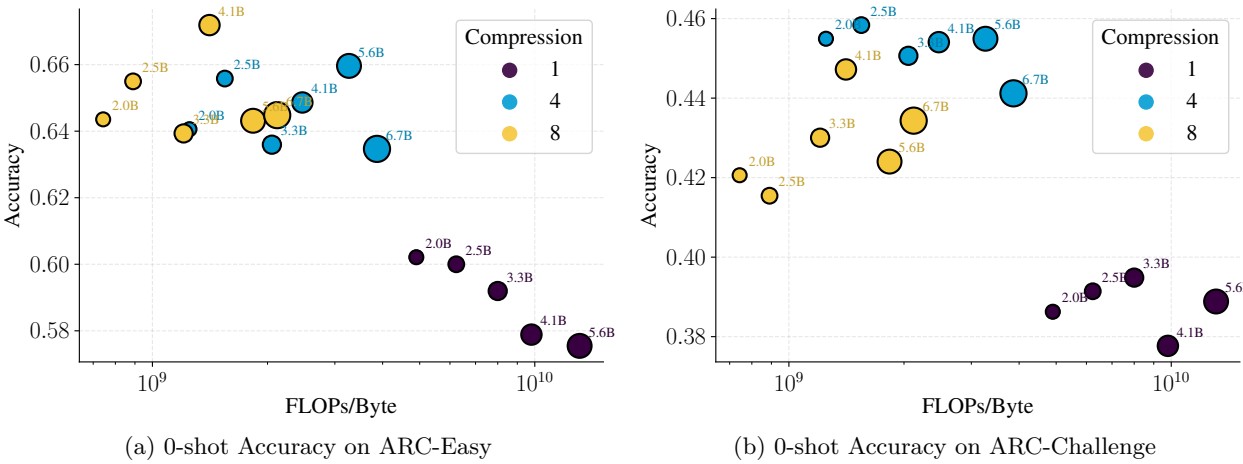

(a) 0-shot Accuracy on ARC-Easy      (b) 0-shot Accuracy on ARC-Challenge

Figure 26: Evaluation of the BLT models trained for $C = 2 \times 10^{21}$ FLOPs on AI2 Reasoning Challange benchmark. The size of each point corresponds to the model parameter count. The results are plotted against inference compute cost per byte, which is dependent on model size $N$ and compression rate $T$.

Figure 26 presents evaluations on multiple-choice questions from the AI2 Reasoning Challenge (Clark et al., 2018). Interestingly, we observe that for the easier version of the task, models with compression rate 8 and compression rate 4 achieve similar scores. The higher compression (compression rate 8) even obtains the best score for the 4.1B-parameter model, while being cheaper to run than the corresponding compression rate 4 model. On the harder "challenge" split, we observe a different pattern: compression rate 4 achieves higher scores than compression rate 8. We conclude that the choice of optimal compression can be task-dependent. More-compressed, and thus cheaper, tokenization may be adequate for easier tasks, while harder tasks may benefit from the additional inference compute associated with lower compression. We also note the underperformance of byte-level models, which we attribute to insufficient data seen during pre-training.

