# OpenReview forum: "Compute Optimal Tokenization"
_TMLR — Under review for TMLR_

### Review · Reviewer_v8d3 · 2026-06-21

**Summary Of Contributions:**

This work investigates the effect of compression rates on the compute efficiency of Large Language Models (LLMs). The authors conducted experiments using the LLaMA 3 transformer architecture across various model scales. The contributions are:

1) A systematic empirical scaling study for latent tokenized models, specifically, Byte Latent Transformer (BLT) architectures. Experiments are conducted to study how the optimal Byte-per-Parameter ratio, $\\rho^\\ast$ - the ratio between optimal training data bytes $(B^\ast)$ and optimal number of model parameters $(N^\\ast)$ - changes across varying compute budgets $(C)$, up to $2 \\times 10^{21}$ FLOPs and compression rates $(T)$, up to12.

2) A numerical analysis for BLT architecture is conducted to study how the optimal loss $(L^\\ast)$ changes across varying $T$.

3) A comparative empirical evaluation comparing the scaling behaviors with $T$ and bits-per-byte (BPB) loss profiles of BLT against various traditional subword-tokenization methods (BPE and SuperBPE).

4) An ablation study to analyze how the language choice affects the compute-optimal compression rate and bytes per parameter ratio, with languages chosen across four distinct script families.

**Strengths:** Conceptual Novelty. This is an evolving and highly relevant area of research. The work tackles an interesting premise and it is of the first to challenge the current understanding of scaling laws.

**Weaknesses:** Poor manuscript readability and structure, which hinders the appreciation of the contributions and a clear understanding of the supporting evidence to the claims.

**Audience:**

Yes

**Audience Explanation:**

Researchers on Large Language Models, Efficient Inference, and Tokenization would be interested in this work.

**Claims And Evidence:**

No

**Claims Explanation:**

The claims are as follows:

>1) $\\rho^\\ast$ remains close to constant across variable compute budget and compression rates. Therefore, for generalization to other tokenizers, the compute-optimal ratio is best expressed in bytes to model parameters.

This claim is not fully supported. The term "close to constant" is imprecise, has not been quantified, and the exact values of $\\rho^\\ast$ cannot be read from Figure 3 or Appendix F.1, nor have the authors provided the reader with these values.

>2) At each training compute budget, there is an optimal compression rate $T^\\ast$. Diverging from its value in either direction increases loss.

This claim is supported by the evidence in Figure 4 and 5.

>3) Discovered scaling trends for models with latent tokenization (BLT) hold for models with subword tokenization (BPE, SuperBPE). Additionally, authors claim that experiments show that lower compression is beneficial for training of larger scale models.

This is claim is not fully supported. While Figure 9 suggests that subword tokenization methods followed similar scaling trends as BLT, it is unclear to me what specific evidence suggests the benefit of lower compression for larger scale models.

>4) $\\rho^\\ast$ and $T^\\ast$ vary across different languages and are correlated with parity.

This claim is supported by Figures 12 and 13 and Table 2, which show differences in optimal values for different languages and show that higher parity translates to a higher optimal compression rate. The parity estimation technique is also practical.

**Requested Changes:**

**Comments on Organization and Writing Style:**

I reiterate that I believe this work studies a novel and very relevant problem that would be of interest to large audiences. However, the current organization, structure and writing style hinder its readability. In its current form, this manuscript is an unpleasant read and - in my opinion - unnecessarily prohibits access of a broader range of researchers into this particular field. Specifically:

>1) The "Prior Work" section is placed towards the end of the paper.

I believe this section must be moved to a standard position within (or just after) the introduction. Currently, the introduction does not fully establish the problem under study, nor does it sufficiently establish the current state of the art (i.e., foundational scaling studies, tokenization methods etc). Without the "Prior Work", the experiments exist in a vacuum, the novelty of the proposed methods cannot be assessed and the claims cannot be verified. Additionally, the introduction is missing important citations that the "Prior Work" provides.

> 2) Latent Tokenized Models (such as BLT) are under-theorized.

Latent Tokenized Models are a very recent and evolving paradigm. For this reason I believe it would be beneficial for the paper to answer the following questions: How are the tokens framed (discrete vectors, patches, embeddings etc)? What are they used for and what problem do they solve (eliminate fixed vocabularies, improve data efficiency, lower computational cost)?

>3)  Some definitions are incomplete.

In Eq.7, $F$ and $E$ are undefined. BPE is not defined as "Byte Pair Encoding".

>4) The manuscript contains some typographical errors.

E.g., Page 8 should read "Similar to *the* last Section. Caption on Figure 7 is unclear "algorithms are obtain"...

**Technical Comments and Questions:**

>1) Critical question for Finding 1

I believe it is necessary for the authors to quantify or define the mathematical bounds of "close to constant". Figure 3 shows the values of $\\rho^\\ast$ to be somewhat close together, but do not report specific values. How close is close enough? Since "close to constant" is a core claim, this comment must be addressed.

>2) Non-critical question for Finding 2

Do/Does the author(s) have insight if the U-shaped curve of Figure 5 is still observed for massive compute budgets (such as $10^{23} etc$)?

>3) Critical question for Finding 3

This work claims that "under a high compute budget, the models with 90% and 75% of vocabulary masked...outperform the models with original BPE tokenizers". In Table 10, when comparing lowest BPB, the difference between masked and original model performance is 0.0001 and 0.0013 respectively. It is difficult to accept differences on the order of $\(10^{-4}\)$ and $\(10^{-3}\)$ as deterministic rankings without being evaluated for statistical significance (such as a t-test). Additionally, the table specifically reports lowest BPB and no variance intervals (or evidence of replicated runs), which adds further doubt to the claims.

---

> ### Author Response · Authors · 2026-07-09
> **Response to the Review**
>
> Thank you for your thorough review and constructive technical questions.
>
> # Re: Organization and Writing Style
>
> First of all, we acknowledge your valid concerns about the manuscript’s clarity and structure
> We have updated the manuscript according to the suggestions: (1) we kept “Prior Work” at the end of the paper but added  "Background" subsections in Methodology  providing necessary background and prior work context, and (2) we included a clearer description of Byte Latent Transformers and the BPE tokenizer variants used
> We have also applied the suggested fixes for incomplete definitions (3) and typographical errors (4).
>
>
> The full list of updates is presented in the general response.
>
> # Re: Technical Comments and Questions
>
> We respond to the concerns regarding the evidence for our claims below, ordered by severity.
>
> ## Re Finding 1: “Close to constant” optimal byte per parameter ratio
>
> We appreciate this request for precision. Our claim that the optimal bytes-per-parameter ratio is "close to constant" across compression rates is based on the fitted Scaling Law I (Eq 4)
> We state a hypothesis  that the value of $\beta$ close to 0.5 means that the optimal byte per parameter ratio remains close to constant with change of compression rate T.
>
> We use the wording “close to constant” instead of “constant”, because achieving $\beta=0.5$ is almost never achieved in statistical fit. However, in the obtained fit we observe that 0.5 falls within 95% confidence interval for $\beta$ (see Table 1), which supports the claim that the compute-optimal bytes-per-parameter ratio is constant up to measurement uncertainty as the compression rate changes.
>
> We  note that analogous trend based argument is made to support close to constant relation between parameters and tokens in Chinchilla compute optimal scaling (Hoffman et al. 2022).
>
> ## Re Finding 3: Optimal compression decreases with scale for subword models
>
> The claim about the change in compute-optimal compression rate is also based on the statistical fit of Scaling Law II (Equation 7).The crucial quantity is the fitted variable $\deltat$, we observe $\deltat$ is positive thus the value of optimal compression rate decreases with the rise of compute (C).
> We have now included this value together with its confidence interval in the summary Table 1 for better visibility.
>
> We agree with the reviewer that the absolute differences between BPE variants in Table 10 are small. We include these results primarily as visual illustrations of the loss achieved by models across compute budgets and compression values (low compression settings exhibit steeper loss decrease with raise in compute than high compression ones). However, we focus on analyzing trends and thus  base the claim on the Scaling Law II fit rather than comparison in Table 10.
> We have clarified this distinction in the updated manuscript.
>
>
> ## Re Finding 2: Extrapolation for larger scales
>
> We expect the "U-shaped" IsoFLOP curves (loss vs. compression) to persist across all scales. While we do not run the full experimental grid beyond 2 x 10^21 FLOPs (please note these experiments are especially expensive due to the need to evaluate the whole grid), we evaluate extrapolation accuracy in Appendix B.3.
>
> There, we have constrained the fitting data for Scaling Law II to 10^21 FLOPs and below and and evaluate extrapolation error for unseen 2 x 10^21 FLOPs models. This experiment shows that “U-shaped” compute dependent loss model (Equation 7) gives more accurate loss prediction than simple “irreducible term” model (Equation 8). RMSE=0.0086 for the former and RMSE=0.0260 for the latter. While this does not directly answer the reviewer's question about scales of 10^23 FLOPs,, this experiment demonstrates that the proposed Scaling Law II provides faithful extrapolation of loss to unseen computation budgets.

---

### Review · Reviewer_jFzT · 2026-06-28

**Summary Of Contributions:**

This paper studies how the compression rate of a tokenizer, measured as average bytes per token, affects compute-optimal scaling for language models. The main claim is that the standard “tokens per parameter” view of scaling is not robust across tokenizers: when compression changes, the more stable quantity is bytes of training data per model parameter. The authors support this by training a large suite of latent-tokenized BLT models across several compression rates and compute budgets, and then fitting scaling laws over model size, training bytes, compression rate, and loss.

The paper’s central findings are: first, the compute-optimal bytes-per-parameter ratio is approximately constant across compression rates, around 60 bytes per parameter for English in their setting. Second, there is a non-monotonic relationship between compression rate and loss: for a fixed compute budget, there is an optimal compression rate, and moving either toward too-fine or too-coarse tokenization hurts performance. Third, this optimal compression rate appears to decrease slowly with increased training compute. Fourth, the same qualitative trends are observed for subword-tokenized models, not only latent-tokenized models. Finally, the authors extend the analysis to several non-English languages and argue that optimal compression and bytes-per-parameter ratios vary by language and correlate with cross-lingual byte parity.

The strongest aspects of the paper are the clear research question, the unusually large and systematic experimental grid, and the attempt to disentangle tokenizer compression from vocabulary size using latent tokenization. The paper also does a good job connecting the results to practical model training decisions, especially the interpretation of Chinchilla-style scaling rules in bytes rather than tokens.

**Audience:**

Yes

**Audience Explanation:**

This paper should be of interest to a broad part of the TMLR audience working on language modeling, scaling laws, tokenization, multilingual modeling, and efficient training/inference. Tokenization is often treated as an implementation detail, while scaling-law work typically expresses data in tokens under a fixed tokenizer. This paper shows that this convention can be misleading when comparing models with different tokenizers, and gives a concrete alternative rule based on training bytes.

The practical implications are significant. The results suggest that model developers should be careful when transferring compute-optimal recipes across tokenizers, vocabularies, or languages, because the same number of tokens may correspond to very different amounts of information. The finding that optimal compression is non-monotonic is also important: it challenges the simple intuition that more compression is always better because it reduces sequence length. The multilingual results are particularly relevant for fairness and efficiency, since they suggest that common multilingual tokenizers may over-compress some languages and under-compress others relative to compute-optimal values.

Even if some details change at larger scales or with different architectures, the conceptual point that scaling laws should account for token information density is valuable and likely to influence future work.

**Broader Impact Concerns:**

No concerns to me.

**Claims And Evidence:**

Yes

**Claims Explanation:**

Overall, I found the evidence convincing. The core empirical design is strong: the authors train many models over a grid of compute budgets, parameter counts, and compression rates, and evaluate using bits-per-byte, which is the appropriate metric for comparing tokenizers with different granularity. The BLT setup is particularly useful because it allows compression rate to be varied more directly than in standard subword tokenization, where compression is entangled with vocabulary size and tokenizer training.

The main claim that bytes-per-parameter is more stable than tokens-per-parameter is well supported by the IsoFLOP plots and the fitted exponents. The fitted values of the compression and compute exponents being close to 0.5 make the interpretation intuitive: as compression changes, compute-optimal configurations preserve a roughly constant ratio of bytes to parameters. The authors also provide several views of the same result, including 2D and 3D IsoFLOP analyses, latent-tokenized and subword-tokenized models, and supplementary plots across compute budgets.

The evidence for an optimal compression rate is also persuasive, though slightly less definitive than the bytes-per-parameter claim. The loss curves clearly look non-monotonic in compression rate, and the proposed residual model improves extrapolation relative to simpler alternatives. However, the precise estimate of the optimal compression rate seems more sensitive to architecture, tokenizer family, task, and fitting assumptions. In particular, the subword experiments necessarily compare tokenizers that differ in more than compression alone, and the downstream results suggest that the best compression rate may vary by task difficulty and inference budget. I do not think these caveats undermine the main claims, but they should be stated more explicitly.

The multilingual results are interesting and directionally convincing, but I would treat them as suggestive rather than fully conclusive. The paper studies a limited number of languages, and language, script, data quality, corpus composition, and parity are hard to disentangle. Still, the artificially inflated English experiment is a useful sanity check, and the observed correlations with parity are plausible and relevant.

**Requested Changes:**

1. The paper sometimes reads as if there is a general optimal compression rate, but the results show that it depends on compute budget, language, tokenization method, and possibly downstream task. The ARC and HellaSwag results suggest that inference-time and task-specific optima can differ from BPB optima. I recommend making this distinction more explicit in the abstract, introduction, and discussion.

2. I would suggest adding more discussion of confounders in the subword experiments. The latent-tokenized results cleanly vary compression, but the subword experiments compare character tokenization, masked BPE variants, BPE, and SuperBPE. These differ in vocabulary, segmentation behavior, token semantics, and perhaps optimization properties, not only in compression rate. The paper acknowledges some of this, but I would like a clearer statement of what the subword experiments do and do not prove.

3. Scaling Law I uses latent/global parameters, excluding encoder/decoder or embedding parameters, while Scaling Law II uses total compute. This may be reasonable, but it is a central modeling choice. Please add a clearer explanation in the main text and, ideally, include a robustness check showing how the main fitted exponents change when total parameters are used instead.

4. The paper reports confidence intervals for some fitted parameters, but the reader would benefit from uncertainty estimates for derived quantities such as the optimal compression rate T, bytes-per-parameter ratio ρ, and language-specific optima. This is especially important for the multilingual section, where conclusions are drawn from relatively few languages.

---

> ### Author Response · Authors · 2026-07-09
> **Response to the Review**
>
> Thank you for your thorough and insightful review.
>
> We appreciate suggestion 1 and have revised the wording throughout to state more clearly that the optimal compression rate is task and dataset dependent.
>
> We have added the “Background” sub-section after “Introduction” offering more details about BPE and its alternative variants used in this work. (suggestion 2). One note regarding confounders in the subword experiments: we note that the masked vocabulary variants exhibit the same segmentation behavior as the original BPE, since they use the same base vocabulary restricted to the 10% or 25% most frequent entries, respectively. Thus, these variants differ primarily in compression rate rather than in segmentation properties. We added this notion in the newly added “Background: Subword Tokenization” subsection.
>
>
> Regarding the compute definition in Scaling Law I and II (suggestion 3), we fitted coefficients $\alpha$ and $\beta$ based on total compute and obtained values close to 0.5: 0.535 and 0.450 respectively, when for an optimal parameter model  (as defined in Equation 1). Although without assumption of $6ND$ compute, we cannot obtain a trend for optimal byte per param fit from this fit.
>
>
> In response to suggestion 4, we have extended Table 1 to include all fitted variables from Scaling Law II, together with corresponding confidence intervals.
>
>
> A detailed list of all manuscript updates is provided in the general response.

---

### Review · Reviewer_oPNn · 2026-06-29

**Summary Of Contributions:**

This paper is a methodical investigation into the effects of data compression on scaling trends in large language models. Rather than expressing data size or compression in terms of tokens, the authors look at data size in terms of actual bytes. The study attempts to determine a set of rules for practical considerations when training large language models. Specifically, they look at i) how the training data size requirements scale with the size of the model, and ii) how the compression rate affects learning given a fixed computational budget.

The paper is thorough and methodical and the authors peform a large slew of experiments to provide support for their claims. The results in the paper are interesting and do lend to the credibility of the authors' claims.

However, the main weakness of the paper is that it was very difficult to follow.

**Audience:**

Yes

**Audience Explanation:**

I think the results of the paper would be of interest to researchers looking for practical considerations when training large language models. The results may provide some justifications for design choices made by practitioners.

However, I do not think that I am the target audience for this paper and I may be missing nuances that someone with more familiarity would be able to see.

**Broader Impact Concerns:**

N/A.

**Claims And Evidence:**

Yes

**Claims Explanation:**

The paper was a practical investigation on the effects of data size and data compression on training language models and understanding how different choices made for training these models influence their performance. The main claim for the paper centers on how best to decide the amount of data needed to train a model. Standard practice conventionally uses the ratio of tokens to model parameters as a metric for this decision. The authors instead propose using the ratio agains the actual data size (in bytes) as the metric regardless of the tokenization used. The first set of results support the hypothesis.

There were three other main claims made by the paper and the results of the paper also agree with the claims made. The experiments were appropriately designed, and the authors were thorough in their comparisons of the different model types, sizes, and computational budgets.

**Requested Changes:**

While the results of paper are interesting, I do not believe the paper is written well enough for publication. It is difficult to read and hard to retain what is being stated. I will outline a few issues that I think can/should be changed.
- The paper lacks a good flow. The authors tried to write the sections out in the order that the research questions were presented, but I felt a lack of flow between the sections.
- The lack of flow was probably exacerbated by the sheer number of acronyms and notations and terms used (B, BLT, BPE, BPB, etc.) It was all given up-front almost immediately and made the paper hard to read. Perhaps structuring the paper to introduce things gradually would make the paper clearer, the flow easier, and make it easier for the reader to see the progression in the questions being asked/answered.
- There are a few formatting issues with the plots e.g., missing axis labels, cut off titles, etc.
- The authors also refer to the appendix in the main results. If results in appendix need to be discussed in the main body they should not be in the appendix. The results in the appendix should purely be supplemental.

---

> ### Author Response · Authors · 2026-07-09
> **Response to the Review**
>
> Thank you for your feedback. We acknowledge the need to clarify definitions and key concepts upfront.
> We have addressed this concern by adding a "Background" subsection after “Introduction” and improving the transitions between sections (suggestions 1 and 2). These changes introduce concepts and notation gradually.
>
> Regarding formatting issues (suggestion 3), we have fixed the cut-off y-axis labels in Figures 4 and 8. Additionally, we have moved the most important results from the appendix into the main text where feasible without extending the main text too much (suggestion 4).
>
> A detailed list of all manuscript updates is provided in the general response.

---

### Author Response · Authors · 2026-07-09
**General Response to Reviews**

We are grateful to all reviewers for their constructive feedback and overall positive assessment of our submission. We are especially thankful for the high evaluation of the adequacy and thoroughness of our experiments: “experiments were appropriately designed, and the authors were thorough in their comparisons” (oPNn), and the praise for a “clear research question, the unusually large and systematic experimental grid” (jFzT).
Moreover, all reviewers agreed that the findings will be of interest to the TMLR audience, with reviewer v8d3 noting that: “The work tackles an interesting premise and it is of the first to challenge the current understanding of scaling laws” and reviewer oPNn stating that the: “paper would be of interest to researchers looking for practical considerations when training large language models”

# Re: Improving Clarity


The recurring concern raised by two reviewers (oPNn, v8d3) pertains to the clarity and organization of the manuscript. We acknowledge that the methodology section was too information-dense and lacked a dedicated prior work section (v8d3), with definitions "given up-front almost immediately" (oPNn). To address these concerns, we have added explanatory “Background” sub-sections in Methodology presenting the key concepts and explaining acronyms used in the paper, e.g.: BLT, BPE, variables: $C$, $N$, $B$, $\rho$.
We also have moved some of the results to the main text to reduce reliance on the appendix, while keeping the  length in check.

# Updates in Paper Revision

We have made the following specific changes to the manuscript to address the concerns of the reviewers.. The updated version is around 2 pages longer, as agreed with the action editor.

- We add “Background” sub-sections clarifying key concepts and definition iin “Methodology” to further describe prior work methods we build on (re: v8d3), specifically we add:
- Subsection describing standard Byte Pair Encoding (BPE) tokenizers along with alternatives (vocabulary masking and  SuperBPE). (re: jFzT)
- Subsection outlining hierarchical language models, latent tokenization and Byte Latent Transformer (BLT). (re: v8d3)
- Subsection defining variables used in compute optimal scaling throughout the paper (re: oPNn)
- We extended Table 1 to include the fitted values and confidence intervals for remaining variables from Scaling Law II.
- Especially the ones governing the trend for optimal compression rate ($T_0$ and $delta$). (re: jFzT, v8d3)
- Improved transitions between sections to enhance readability and flow. (re: oPNn)
- Fixed the cut-off y-axis labels in Figures 4 and 8 (re: v8d3)
- Incorporated text improvements and clarifications  to reflect suggestions from reviewers. (re: jFzT, oPNn, v8d3)



We address concerns regarding methodology and specific experiments in our individual responses to each reviewer below.